# Suppression of transcytosis regulates zebrafish blood-brain barrier function

Natasha M O'Brown[1], Sean G Megason[2]*, Chenghua Gu[1]*

[1]Department of Neurobiology, Harvard Medical School, Boston, United States;
[2]Department of Systems Biology, Harvard Medical School, Boston, United States

**Abstract** As an optically transparent model organism with an endothelial blood-brain barrier (BBB), zebrafish offer a powerful tool to study the vertebrate BBB. However, the precise developmental profile of functional zebrafish BBB acquisition and the subcellular and molecular mechanisms governing the zebrafish BBB remain poorly characterized. Here, we capture the dynamics of developmental BBB leakage using live imaging, revealing a combination of steady accumulation in the parenchyma and sporadic bursts of tracer leakage. Electron microscopy studies further reveal high levels of transcytosis in brain endothelium early in development that are suppressed later. The timing of this suppression of transcytosis coincides with the establishment of BBB function. Finally, we demonstrate a key mammalian BBB regulator Mfsd2a, which inhibits transcytosis, plays a conserved role in zebrafish, as *mfsd2aa* mutants display increased BBB permeability due to increased transcytosis. Our findings indicate a conserved developmental program of barrier acquisition between zebrafish and mice.

DOI: https://doi.org/10.7554/eLife.47326.001

*For correspondence:
Sean_Megason@hms.harvard.edu
(SGM);
Chenghua_Gu@hms.harvard.edu
(CG)

**Competing interests:** The
authors declare that no
competing interests exist.

**Reviewing editor:** Richard
Daneman, University of
California, San Diego, United
States

## Introduction

Blood vessels in the vertebrate brain are composed of a single layer of endothelial cells that possess distinct functional properties that allow the passage of necessary nutrients yet prevent unwanted entry of specific toxins and pathogens into the brain. This specialized endothelial layer forms the blood-brain barrier (BBB) and restricts the passage of substances between the blood and the brain parenchyma via two primary mechanisms: 1) specialized tight junction complexes between apposed endothelial cells to prevent intercellular transit (*Reese and Karnovsky, 1967*; *Brightman and Reese, 1969*) and 2) suppressing vesicular trafficking or transcytosis to prevent transcellular transit (*Ben-Zvi et al., 2014*; *Andreone et al., 2015*; *Chow and Gu, 2017*). BBB selectivity is further refined with the expression of substrate-specific transporters that dynamically regulate the influx of necessary nutrients and efflux of metabolic waste products (*Sanchez-Covarrubias et al., 2014*; *Umans et al., 2017*). While the BBB is comprised of endothelial cells, the surrounding perivascular cells including pericytes and astroglial cells, play a critical role in forming and maintaining barrier properties (*Janzer and Raff, 1987*; *Armulik et al., 2010*; *Bell et al., 2010*; *Daneman et al., 2010*; *Wang et al., 2014*). Collectively, endothelial cells and the surrounding perivascular cells form the neurovascular unit.

As the simplest genetic model organism with an endothelial BBB (*Jeong et al., 2008*), zebrafish offer a powerful tool to study the cellular and molecular properties of the vertebrate BBB (*Xie et al., 2010*; *Vanhollebeke et al., 2015*; *Umans et al., 2017*; *O'Brown et al., 2018*; *Quiñonez-Silvero et al., 2019*). Zebrafish have served as a great model system to study vascular biology due to their large clutch size, rapid and external development, and transparency for in vivo whole organism live-imaging (*Lawson and Weinstein, 2002*; *Jin et al., 2005*; *Santoro et al., 2007*; *Armer et al., 2009*; *Herbert et al., 2009*; *Phng et al., 2009*; *Geudens et al., 2010*; *Herbert et al., 2012*; *Wilkinson and van Eeden, 2014*; *Franco et al., 2015*; *Vanhollebeke et al., 2015*;

*Matsuoka et al., 2016*; *Ulrich et al., 2016*; *Venero Galanternik et al., 2017*; *Stratman et al., 2017*; *Geudens et al., 2018*). Additionally, with the advent of CRISPR-Cas9 technology, zebrafish provide an efficient genetic toolkit for targeted mutagenesis (*Hwang et al., 2013*; *Gagnon et al., 2014*; *Ablain et al., 2015*; *Varshney et al., 2015*; *Albadri et al., 2017*; *Hogan and Schulte-Merker, 2017*). However, the subcellular and molecular mechanisms governing the formation and maintenance of the zebrafish BBB are only beginning to be characterized. Expanding our understanding of the zebrafish BBB can thus reveal the mechanistic similarities between the zebrafish and mammalian BBB to further evaluate the position of zebrafish as a model organism for studying the BBB.

Barrier properties of brain endothelial cells are induced by extrinsic signals from other cells in the surrounding microenvironment during development (*Stewart and Wiley, 1981*). In rodents, the BBB becomes functionally sealed in a spatiotemporal gradient, with the hindbrain and midbrain barriers becoming functional before the cortical barrier (*Daneman et al., 2010*; *Ben-Zvi et al., 2014*; *Sohet et al., 2015*). Within the cortex, barrier function is acquired along a ventral-lateral to dorsal-medial gradient (*Ben-Zvi et al., 2014*). In the zebrafish, existing studies have disagreed over the timing of zebrafish barrier formation, with some suggesting that BBB maturation occurs concurrently with angiogenesis (*Jeong et al., 2008*; *Xie et al., 2010*; *Umans et al., 2017*) and others providing a wide range beginning at 3 dpf and extending to 10 dpf (*Fleming et al., 2013*). These conflicting reports may be due to regional developmental gradients of barrier acquisition, differences in definitions of BBB development, which range from functional tracer restriction (*Jeong et al., 2008*; *Fleming et al., 2013*) to gene expression of BBB-specific selective transporters, such as *glut1b* (*Umans et al., 2017*), or variations in the experimental approaches used to assess BBB permeability such as the molecular weight and properties of tracers and circulation time. Therefore, we aim to resolve some of these discrepancies through detailed regional and subcellular characterizations of functional barrier acquisition in zebrafish.

Recent work in the mammalian blood-retinal barrier has indicated that the suppression of transcytosis governs functional barrier development (*Chow and Gu, 2017*). Interestingly, endothelial cells at the leaky neonatal angiogenic front possess functional tight junction complexes halting the intercellular passage of the tracer protein Horseradish Peroxidase (HRP) at the so-called 'kissing points'. In contrast, these endothelial cells exhibit high levels of HRP-filled vesicles compared to functionally sealed proximal vessels. Moreover, these areas of elevated vesicular trafficking continue to correspond with barrier permeability at the angiogenic front until the barrier seals (*Chow and Gu, 2017*). Work in the mouse BBB has also demonstrated the importance of suppressing transcytosis in determining barrier permeability. Mice lacking the major facilitator super family domain containing 2a (Mfsd2a) lipid transporter exhibit increased levels of caveolae vesicles in CNS endothelial cells, resulting in increased barrier permeability (*Ben-Zvi et al., 2014*; *Andreone et al., 2017*). Whether this suppression of transcytosis also determines BBB function in zebrafish remains unknown.

Here in zebrafish, we find a spatiotemporal gradient of barrier acquisition, and capture the dynamics of BBB leakage as it matures during development using time lapse live imaging. We further find a conserved role for transcytosis suppression in determining barrier function, both during normal development and in *mfsd2aa* mutants.

## Results

### Posterior-anterior gradient of zebrafish BBB development

To determine when and how the zebrafish BBB becomes functional in different brain regions, we performed intracardiac injections of fluorescently conjugated tracers (1 kDa NHS and 10 kDa Dextran) simultaneously at different developmental stages and imaged live fish after 1 hr of tracer circulation (*Figure 1A and B*). NHS is widely used to assess mouse BBB permeability (*Sohet et al., 2015*; *Chow and Gu, 2017*; *O'Brown et al., 2018*). Additionally, NHS has been used successfully to assess junctional defects in occludin- and claudin-deficient animals (*Chen et al., 1997*; *Furuse et al., 2002*; *Nitta et al., 2003*), and was previously shown to be restricted within the adult zebrafish cerebral vasculature (*Jeong et al., 2008*). We used a combination of different molecular weight tracers to tease apart potential avenues of leakage, as tight junctional defects result specifically in the leakage of low-molecular-weight tracers 1 kDa and below into the brain parenchyma (*Nitta et al., 2003*; *Campbell et al., 2008*; *Sohet et al., 2015*; *Yanagida et al., 2017*). To assess BBB permeability, we

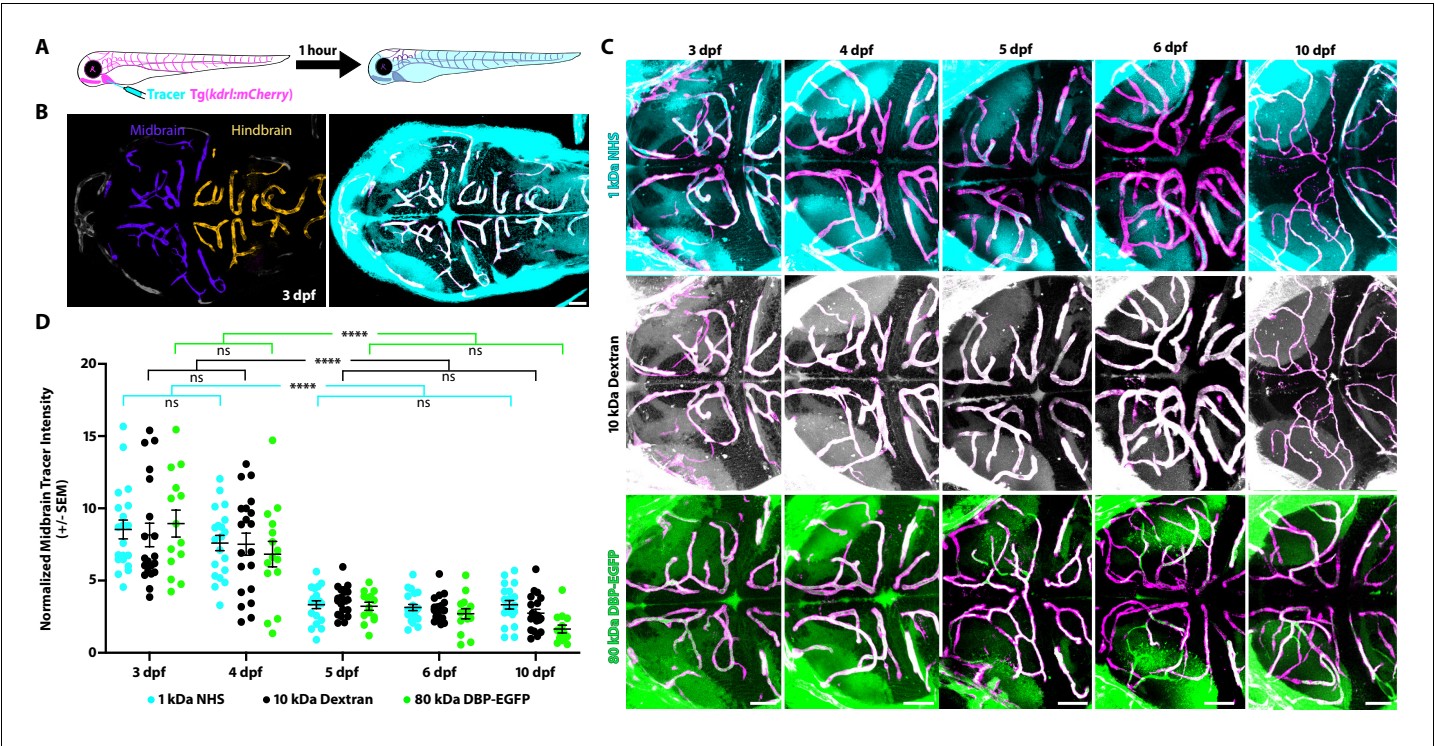

**Figure 1.** The midbrain BBB becomes functional at 5 dpf. (**A**) Diagram of the tracer leakage assay. Fluorescently conjugated tracers (turquoise) were injected intracardially into transgenic fish that express mCherry in the vasculature (magenta; Tg(*kdrl:mCherry*)) and allowed to circulate for 1 hr before imaging. (**B**) Dorsal view maximum intensity projection of the larval brain vasculature at 3 dpf. Left image is pseudo-colored to demarcate the midbrain (violet) and the hindbrain (gold) vasculature. Right image shows the NHS tracer (turquoise) in the entire larval brain, with a large number of tracer-filled parenchymal cells in the midbrain. (**C**) Representative dorsal view maximum intensity projections of larval zebrafish midbrains at different developmental stages reveal increased permeability at 3 and 4 dpf compared to 5, 6 and 10 dpf. The increased early permeability was observed with two injected tracers of different sizes, a 1 kDa NHS (turquoise) and a 10 kDa Dextran (white), as well with an 80 kDa transgenic serum protein DBP-EGFP (green). Parenchymal tracer intensity outside of the vasculature (magenta) was measured and normalized to the blood vessel tracer intensity in each fish. Scale bars represent 50 µm. (**D**) Quantification of normalized parenchymal tracer intensity in the midbrain between 3 and 10 dpf reveals a significant decrease in tracer leakage at five dpf. There was no difference observed between different tracers at any time point. There was no significant change from 3 to 4 dpf or from 5 to 10 dpf, suggesting that the midbrain barrier seals around 5 dpf. N = 14–21 fish, each represented as a single dot on the plot. The mean and the standard error are drawn in black for each tracer and stage. ****p<0.0001, ns is not significant by two-way ANOVA.

DOI: https://doi.org/10.7554/eLife.47326.002

The following source data and figure supplement are available for figure 1:

**Source data 1.** The midbrain BBB becomes functional at 5 dpf and the hindbrain BBB becomes functional at 4 dpf.

DOI: https://doi.org/10.7554/eLife.47326.004

**Figure supplement 1.** The hindbrain BBB becomes functional at 4 dpf.

DOI: https://doi.org/10.7554/eLife.47326.003

measured tracer fluorescence intensity in the brain parenchyma and normalized to circulating levels of each tracer, that is to the fluorescence intensity of tracer within brain blood vessels, to account for between fish variation in tracer injections or circulation (details in the Materials and methods section). At 3 dpf, we observed the presence of both NHS and Dextran throughout the brain parenchyma (average of 8.5 ± 0.3 Tracer Intensity in the midbrain and average of 7.4 ± 0.4 NHS and 5.5 ± 0.6 Dextran Intensity in the hindbrain; *Figure 1*; *Figure 1—figure supplement 1*). These leakage assays revealed that the injected tracers were able to permeate into the brain parenchyma, suggesting that the BBB was not functionally sealed. In addition to the use of exogenous injected fluorescent tracers, we also assayed BBB permeability with an endogenous transgenic serum DBP-EGFP fusion protein (Tg(*l-fabp:DBP-EGFP*)) to account for injection artifacts (*Xie et al., 2010*). At 3 dpf, we observed similar leakage patterns with the transgenic serum protein as we did with the injected tracers (average of 8.9 ± 0.8 DBP-EGFP Intensity in the midbrain and average of 6.1 ± 0.4 DBP-EGFP Intensity in the hindbrain; *Figure 1C and D*; *Figure 1—figure supplement 1*). At 4 dpf,

we observed a significant decrease in tracer intensity in the hindbrain (average of 4.1 ± 0.2 NHS Intensity, 3.9 ± 0.4 Dextran Intensity and 4.0 ± 0.4 DBP-EGFP Intensity; p<0.0001, p=0.0115 and p=0.0034 compared to 3 dpf for NHS, Dextran and DBP-EGFP by two-way ANOVA; *Figure 1—figure supplement 1*). However, the midbrain barrier remains leaky (average of 7.6 ± 0.5 NHS Intensity, 7.5 ± 0.8 Dextran Intensity and 6.8 ± 0.9 DBP-EGFP Intensity; p=0.66, p=0.89 and p=0.079 compared to 3 dpf for NHS, Dextran and DBP-EGFP by two-way ANOVA; *Figure 1C and D*). At 5 dpf, the midbrain parenchymal tracer intensities are dramatically reduced (average of 3.3 ± 0.3 NHS Intensity, 3.6 ± 0.2 Dextran Intensity and 3.2 ± 0.3 DBP-EGFP Intensity; p<0.0001, two-way ANOVA; *Figure 1B and C*) and no significant changes were observed in the hindbrain (*Figure 1—figure supplement 1*). No significant change was observed in midbrain tracer uptake from 5 to 6 dpf or from 5 to 10 dpf (*Figure 1C and D*), indicating that the midbrain barrier becomes sealed at 5 dpf. Of note, all three tracers showed nearly indistinguishable patterns of uptake, with similar parenchymal tracer intensity levels (*Figure 1*; *Figure 1—figure supplement 1*), suggesting that the leakage is not due to tight junctional defects but may rather be due to an increase in vesicular trafficking.

## Time lapse imaging reveals two modes of developmental leakage

The vast majority of studies interrogating BBB permeability have relied on observing leakage in fixed static images from mutant mice, which does not reveal the dynamic nature of the BBB. A key advantage of zebrafish is the ability to examine biological processes in vivo in real-time, providing us with the unique opportunity to observe the process of functional barrier maturation (*Figure 1*). To examine the developmental dynamics of barrier leakage, we injected larvae at 3 dpf with fluorescently labeled 10 kDa Dextran and performed time lapse live imaging for 1 hr following injection, with time 0 being an average of 8 min post-injection. We observed a gradual and diffuse increase in extravascular Dextran intensity over time within the brain parenchyma (slope of 0.13 intensity/minute; *Figure 2—videos 1* and *2*; *Figure 2A, C and E*). In addition to the time-dependent increase in overall Dextran intensity in the brain parenchyma, we also observed parenchymal cells taking up tracer, both directly adjacent to blood vessels and at a small distance away (*Figure 2A and C*). To validate that the Dextran was in fact being taken up by cells rather than filling the extracellular space surrounding cells, we performed these same tracer injections in a transgenic line with fluorescently labeled cell membranes (Tg(*actb2:MA-Citrine*)) at 3 dpf and observed several tracer-filled cells, that is tracer confined within cell membranes outside of the vasculature (*Figure 2—figure supplement 1*). Similarly, when we performed these Dextran injections in a nuclear labeled transgenic line (Tg(*actb2:Hsa.H2B-Scarlet*)) at 3 dpf, we also observed Dextran co-localization with nuclear signal (*Figure 2—figure supplement 1*), further indicating that the injected tracers are taken up by scavenger parenchymal cells in addition to permeating the extracellular spaces.

Interestingly, at 3 dpf, we observed an actively sprouting cell that extended away from the established vessel in a sporadic fashion (*Figure 2—Video 2*; *Figure 2A*). After 20 min of rambling migration, this sprout suddenly released a large bolus of Dextran that appeared to be taken up by a single parenchymal cell. Two min later, this same sprout released a second bolus of Dextran on the opposite side (*Figure 2—Video 2*). These rare large bursts of leakage were not unique to this sprout; they were also sporadically observed from established blood vessels (*Figure 2—Video 1*; *Figure 2A*). While this particular fish displayed three of these bursting events, defined as parenchymal Dextran intensity increasing to luminal levels in less than 1 min, only two of the other five measured 3 dpf fish displayed any bursts within the hour post tracer injection, one from a sprouting tip cell and one from a stable vessel, further indicating the infrequency of these large bursts. Cumulatively, this time lapse data revealed two discrete types of leakage occurring during this early developmental stage: steady and diffuse Dextran leakage into the parenchyma that makes up the vast majority of observed leakage and rare large bursts of leakage.

When we performed these same time lapse experiments at 5 dpf, we observed significantly less overall tracer accumulation in the brain parenchyma over the course of the hour in addition to reduced rates of tracer accumulation (slope of 6.833e-6 intensity/sec; *Figure 2—Video 3*; p<0.001, Mann-Whitney test; *Figure 2B, D and E*). Furthermore, 5 dpf fish never exhibited bursts of tracer leakage as observed at 3 dpf. These data suggest that at early stages, most tracer leakage occurs broadly through vessel walls via unrestricted transcytosis with an occasional transient rupture to endothelial integrity that leads to these large bursts of leakage. Once tracer is leaked into the

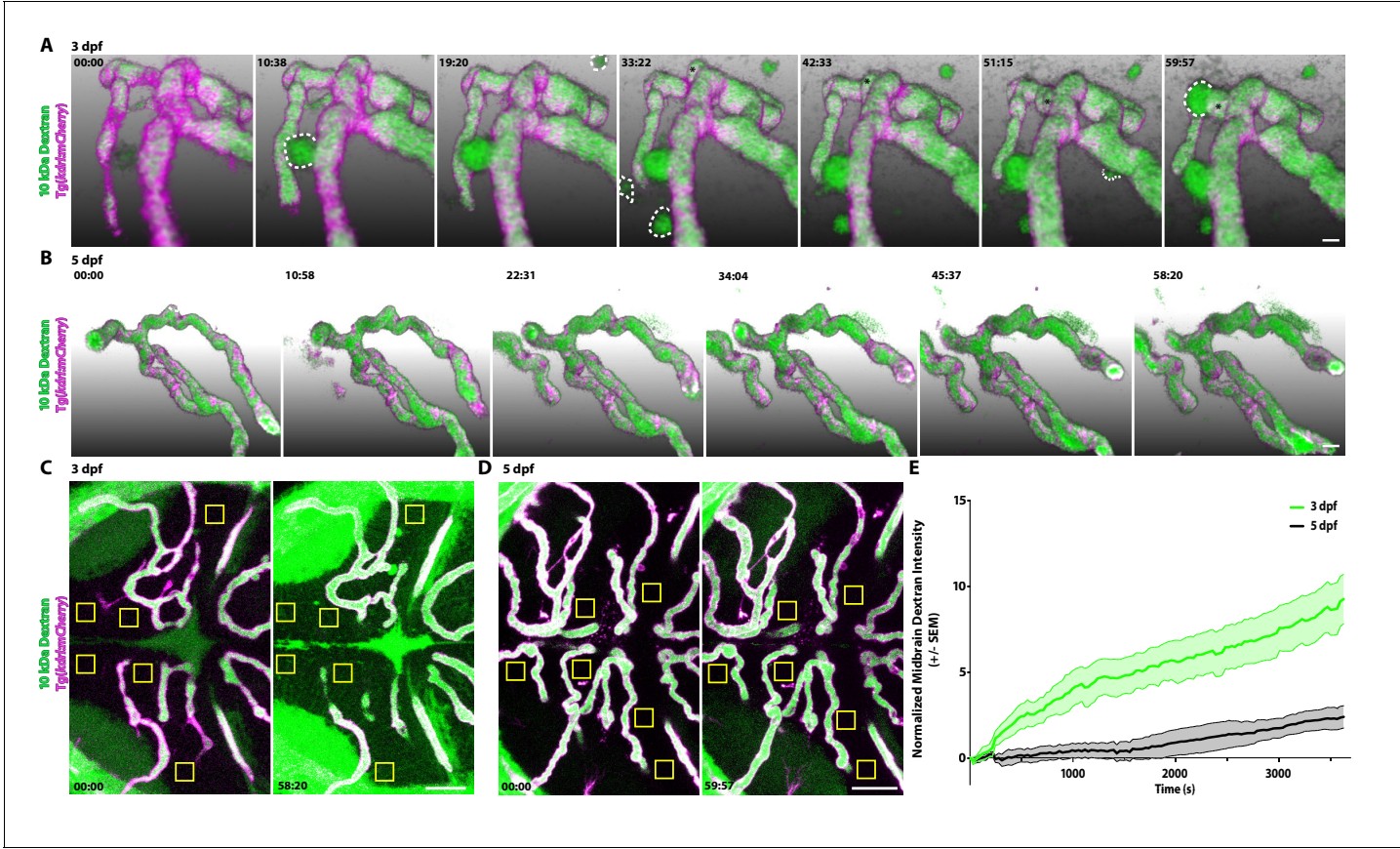

**Figure 2.** Dynamic tracer leakage in the developing BBB via live imaging. (**A**) Time course stills from *Figure 2—Video 1* of tracer leakage at 3 dpf reveal an increase in parenchymal cells absorbing the Dextran tracer (outlined by dashed white lines) as well as a general increase in overall Dextran (green) intensity outside of the vasculature (magenta). An angiogenic tip cell becomes apparent at 33:23 and is demarcated by an asterisk (*). This tip cell produces two separate bursts of leakage observed in *Figure 2—Video 2*. (**B**) Time course stills from *Figure 2—Video 3* of Dextran tracer dynamics at 5 dpf reveals a mature BBB, with reduced overall Dextran extravasation into the brain parenchyma. The scale bars represent 10 µm. (**C and D**) Dorsal maximum intensity projection of the midbrain at 3 dpf (**C**) and 5 dpf (**D**) at the first and last time point examined. While there is a large increase in overall parenchymal Dextran intensity over time at 3 dpf, the 5 dpf midbrain parenchyma appears relatively unaltered after an hour of Dextran circulation. Boxed regions are representative of the six areas per fish used for analysis in E. The scale bars represent 50 µm. (**E**) Quantification of Dextran intensity in the brain parenchyma over time at 3 dpf (green) and 5 dpf (black) shows a significant difference in tracer leakage dynamics (p<0.0001, Mann Whitney U test), with both more total Dextran accumulation and a faster rate of Dextran accumulation in the brain parenchyma at 3 dpf. N = 6 fish with 6 regions analyzed per fish.

DOI: https://doi.org/10.7554/eLife.47326.005

The following video, source data, and figure supplement are available for figure 2:

**Source data 1.** Dynamic tracer leakage in the developing BBB via live imaging.
DOI: https://doi.org/10.7554/eLife.47326.007
**Figure supplement 1.** Injected tracers are taken up by parenchymal cells.
DOI: https://doi.org/10.7554/eLife.47326.006
**Figure 2—video 1.** Time lapse of Dextran leakage into the brain parenchyma at 3 dpf.
DOI: https://doi.org/10.7554/eLife.47326.008
**Figure 2—video 2.** Time lapse of Dextran leakage into the brain parenchyma at 3 dpf.
DOI: https://doi.org/10.7554/eLife.47326.009
**Figure 2—video 3.** Time lapse of Dextran impermeability at 5 dpf.
DOI: https://doi.org/10.7554/eLife.47326.010

intercellular space of the parenchyma it can be taken up by select parenchymal cells (*Figure 2—figure supplement 1*). As the BBB matures both sources of leakage sharply decrease.

## Suppression of transcytosis coincides with functional BBB development

Given the observed differences in brain permeability during larval development, we next sought to determine the subcellular mechanisms underlying the development of a functional BBB. Therefore, we assessed BBB properties by performing intracardiac injections of electron-dense NHS-gold nanoparticles (5 nm) followed by transmission electron microscopy (TEM) at different developmental stages when the BBB is leaky (3 dpf) and when it is functionally sealed to circulating fluorescent tracers (5 and 7 dpf; *Figure 3*). All the blood vessels analyzed had a maximal diameter of 5 µm to enrich for capillaries and small veins. At 3 dpf, we found blood vessels in direct contact with neurons and pericytes, with the pericytes sharing the endothelial basement membrane (*Figure 3A*), as in the

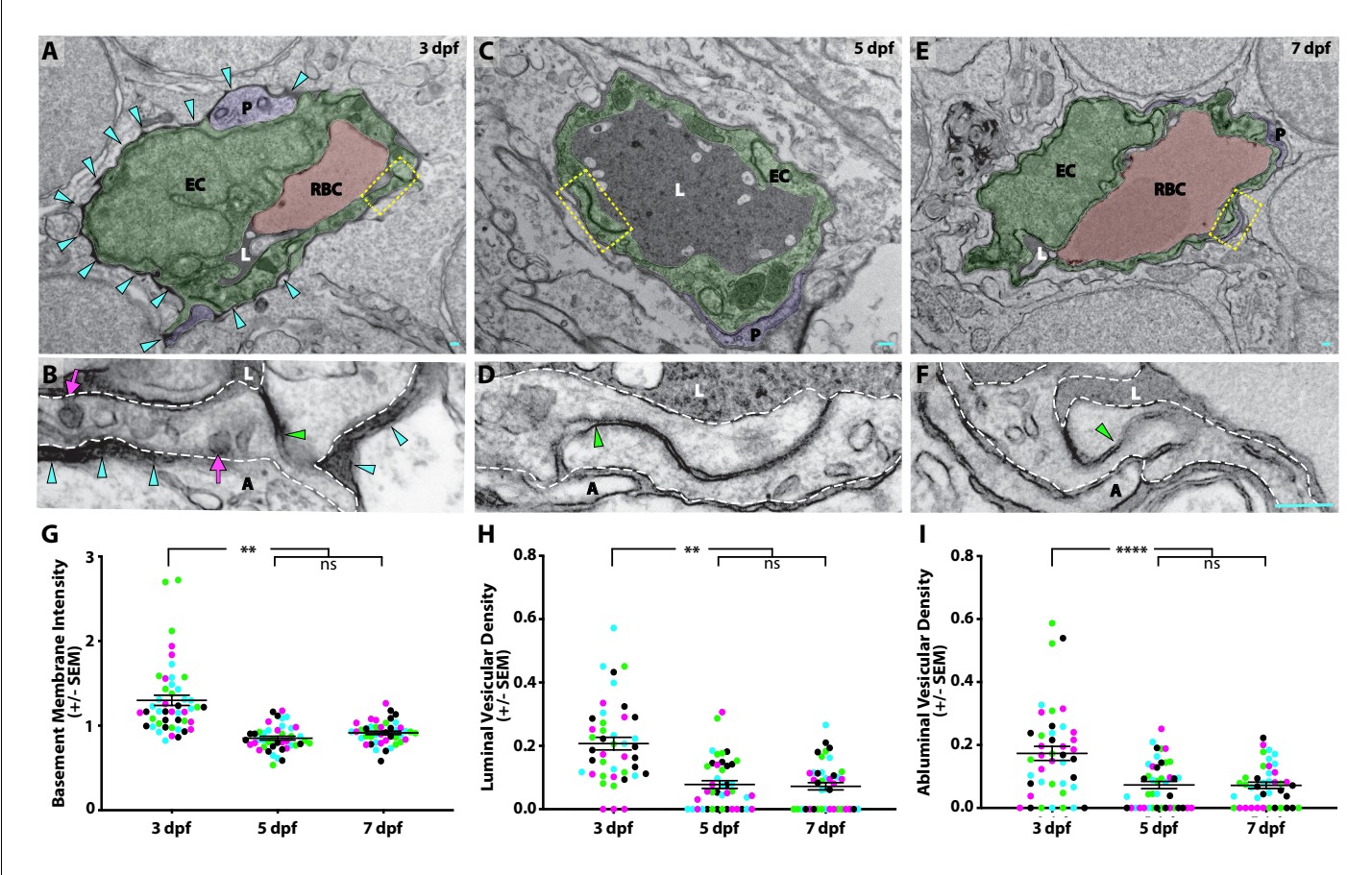

**Figure 3.** Suppression of transcytosis determines the timing of functional BBB formation. (**A, C, E**) TEM images of individual blood vessel cross-sections after injection of electron-dense gold nanoparticles at 3 dpf (**A**), 5 dpf (**C**), and 7 dpf (**E**). Endothelial cells (EC) are pseudo-colored green, pericytes (P) are pseudo-colored purple and red blood cells (RBC) are pseudo-colored red when present in the lumen (L). Turquoise arrowheads highlight the gold-filled basement membrane at 3 dpf (**A**). (**B, D, F**) High magnification images (25000x) of the areas boxed in A, C, and E, respectively, with the endothelial cells outlined with white dashed lines. The images are oriented with the lumen (L) on top and the ablumen (A) on the bottom. Tight junctions are functional as early as 3 dpf (**B**), as seen by their ability to halt the gold nanoparticles at the so-called 'kissing point' (green arrowhead), and remain functional throughout development (**D and F**). Even though the tight junctions are functional at 3 dpf, the endothelial basement membrane is filled with electron-dense gold nanoparticles (B, turquoise arrowheads). This appears to be due to an elevated level of luminal and abluminal gold-filled vesicles (magenta arrows). The scale bars represent 200 nm. (**G**) Quantification of the endothelial basement membrane gold intensity normalized to luminal gold intensity. (**H and I**) Quantification of the vesicular densities both on the luminal (**H**) and abluminal (**I**) membrane of endothelial cells reveals a suppression of vesicular densities beginning at 5 dpf that remains constant at 7 dpf. N = 4 fish, each marked with a different color, with at least 10 blood vessels quantified for each fish and displayed as single points. ****p<0.0001, **p<0.01, ns is not significant by nested one-way ANOVA.
DOI: https://doi.org/10.7554/eLife.47326.011

The following source data is available for figure 3:

**Source data 1.** Quantification of basement membrane gold intensity and vesicular densities during development.
DOI: https://doi.org/10.7554/eLife.47326.012

mammalian neurovascular unit. During this leaky developmental stage, the endothelial basement membrane was filled with the electron-dense gold nanoparticles (*Figure 3A and B*; turquoise arrowheads) with an average gold intensity of $1.29 \pm 0.07$ (luminal gold intensity normalized to 1.0; *Figure 3G*), further demonstrating that the BBB is functionally immature at 3 dpf. To decipher how the gold nanoparticles traverse from the lumen to the basement membrane, we first looked at the tight junctions to see if the nanoparticles passed between apposed endothelial cells. We consistently observed that the nanoparticles were halted at the 'kissing points' between endothelial cells, indicating that tight junctions were functional prior to formation of a functional BBB (49/49 functional tight junctions; *Figure 3B*; green arrowhead). The presence of functional tight junctions at this stage is in agreement with the previously reported expression of tight junction proteins ZO-1 and Claudin-5 in cerebral blood vessels at 3 dpf (*Jeong et al., 2008*; *Xie et al., 2010*). As transcytosis has been implicated in the maturation of the blood-retinal barrier (*Chow and Gu, 2017*), we next examined the levels of luminal and abluminal flask-shaped vesicles filled with gold nanoparticles as a means of assessing transcytosis. Quantification of gold-filled luminal and abluminal flask-shaped vesicles revealed an average of $0.21 \pm 0.03$ and $0.17 \pm 0.02$ vesicles/μm, respectively (*Figure 3H and I*). These data reveal that the basement membrane becomes filled with gold nanoparticles via vesicular transport rather than intercellular passage between immature tight junctions.

When we repeated this assay at 5 dpf, when the barrier becomes less permeable to fluorescently conjugated tracers (*Figures 1* and *2*), we still observe close contacts between endothelial cells and pericytes (*Figure 3C*). At 5 dpf, the basement membrane was noticeably lacking gold particles as compared to 3 dpf (*Figure 3D*), with a reduced average gold intensity of $0.85 \pm 0.02$ (*Figure 3G*). Like at 3 dpf, the tight junctions remained functional at 5 dpf, based on their capacity to halt gold nanoparticles at the kissing points between neighboring endothelial cells (76/76 functional tight junctions; *Figure 3D*). However, the levels of vesicles both luminally and abluminally were notably decreased to $0.08 \pm 0.01$ and $0.07 \pm 0.01$ vesicles/μm, respectively ($p<0.001$ (luminal) and $p<0.0001$ (abluminal), nested two-way ANOVA; *Figure 3H and I*). At 7 dpf, the neurovascular cellular interactions remained constant with endothelial cells in close contact with pericytes and neurons, and with an unfilled basement membrane (average gold intensity of $0.91 \pm 0.02$; *Figure 3E and G*). Importantly, the tight junctions remained functional (51/51 functional tight junctions), and the low vesicular densities observed at 5 dpf remained comparably low at 7 dpf with $0.07 \pm 0.01$ vesicles/μm, both luminally and abluminally (*Figure 3F, H and I*). Taken together with the fluorescent tracer data, our data suggests that the zebrafish BBB becomes functionally impermeable at 5 dpf via the suppression of vesicular trafficking.

## Conserved role of Mfsd2a in determining BBB function

Given the important role of suppressing transcytosis in determining the developmental maturation of the mouse BBB, and a similar correlation of the timing between the suppression of transcytosis and functional barrier formation in zebrafish, we wondered whether the key mammalian barrier regulator Mfsd2a, which suppresses caveolae mediated transcytosis (*Ben-Zvi et al., 2014*; *Andreone et al., 2017*), plays a conserved role in zebrafish. Zebrafish contain two paralogues of *Mfsd2a*, *mfsd2aa* and *mfsd2ab,* both of which are expressed in the developing zebrafish CNS (*Guemez-Gamboa et al., 2015*). In order to resolve the vascular expression of both paralogues, we performed fluorescent in situ hybridization (FISH) in *kdrl:mCherry* transgenic fish sections at 3 and 5 dpf and in adult brain sections. To control for off-target signal, we also performed FISH with a mouse *Gfap* probe, which should not bind to the zebrafish transcript (*Figure 4*). Vascular expression levels for *mfsd2aa* or *mfsd2ab* were then background corrected using the measurements from the *Gfap* probe. As a positive control, we also performed FISH for the BBB-specific vascular marker *slc2a1a* (also known as *glut1b*) (*Umans et al., 2017*; *Quiñonez-Silvero et al., 2019*), and observed consistent *slc2a1a* signal in cerebral vessels at all time points examined (*Figure 4—figure supplement 1*). FISH revealed undetectable levels of *mfsd2aa* expression in cerebral vessels at 3 dpf, higher levels at 5 dpf, when BBB endothelial cells suppress transcytosis (*Figure 3*), and intermediate levels in adult blood vessels (*Figure 4*). *Mfsd2ab*, on the other hand, is highly expressed in cerebral vessels at 3 and 5 dpf and lowly expressed in adult vessels (*Figure 4*). *Mfsd2aa* is 61% identical to human *MFSD2A* and 62% identical to mouse *Mfsd2a* (*Figure 4—figure supplement 2A*). *Mfsd2ab* is 64% identical to human *MFSD2A* and mouse *Mfsd2a* (*Figure 4—figure supplement 2A*). The two paralogues are only 68% identical to each other, but they both contain the lipid binding domain that

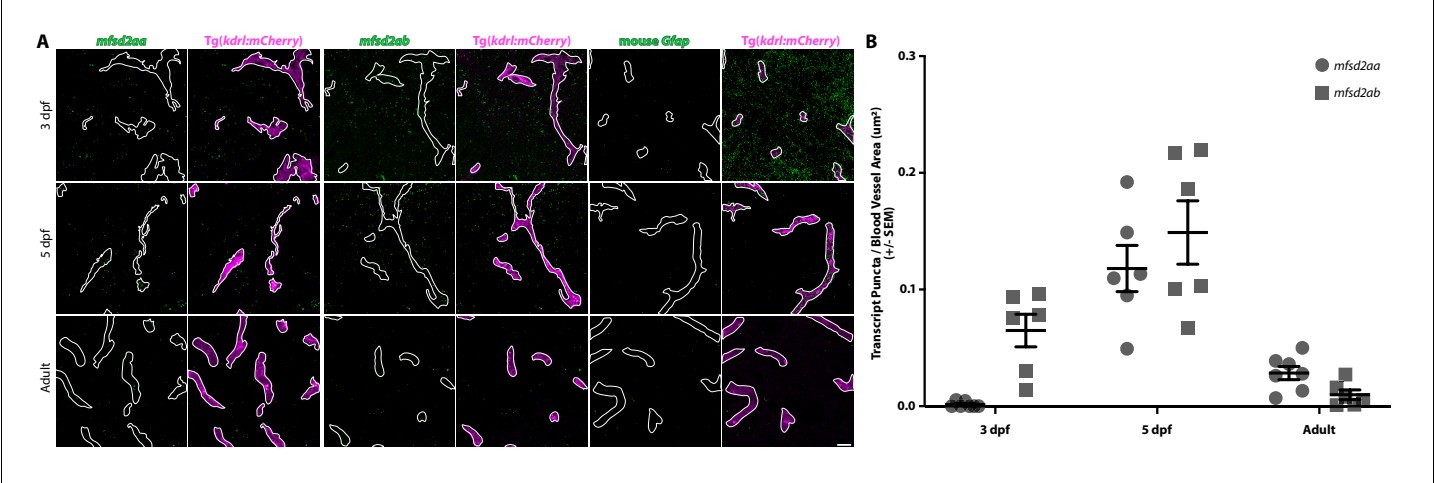

**Figure 4.** Fluorescent in situ hybridization (FISH) reveals vascular signal for both *mfsd2aa* and *mfsd2ab* at 5 dpf. (**A**) FISH for *mfsd2aa* (left), *mfsd2ab* (middle) and mouse *Gfap* negative control (right) in Tg(*kdrl:mCherry*) 3 dpf, 5 dpf and adult brain tissue. FISH for *mfsd2aa* (left) reveals no vascular expression above background at 3 dpf, high levels at 5 dpf and low levels in adult blood vessels. FISH for *mfsd2ab* (middle) reveals high levels of vascular signal at 3 and 5 dpf and negligible vascular signal in adults. Neither *Mfsd2a* paralogue was expressed exclusively in the vasculature, outlined in white, at any time examined and *mfsd2ab* displayed higher overall expression throughout the larval brain than *mfsd2aa*. Scale bar represents 10 μm. (**B**) Quantification of the number of transcript puncta per blood vessel area in 3 dpf, 5 dpf and adult brain sections after background (mouse *Gfap* expression) subtraction. N = 6 sections from at least 3 different fish.

DOI: https://doi.org/10.7554/eLife.47326.013

The following source data and figure supplements are available for figure 4:

**Source data 1.** Fluorescent in situ hybridization (FISH) and qPCR quantification for *mfsd2aa* and *mfsd2ab*.
DOI: https://doi.org/10.7554/eLife.47326.016
**Figure supplement 1.** FISH reveals consistent vascular *slc2a1a* signal throughout BBB development.
DOI: https://doi.org/10.7554/eLife.47326.014
**Figure supplement 2.** Zebrafish have two *Mfsd2a* paralogues, *mfsd2aa* and *mfsd2ab*.
DOI: https://doi.org/10.7554/eLife.47326.015

is critical for governing barrier properties (*Figure 4—figure supplement 2A*; *Andreone et al., 2017*). Given the lack of a clear paralogue that most closely resembles *Mfsd2a*, we generated CRISPR mutants for both paralogues independently. *Mfsd2aa*[hm37/hm37] mutants have a 7 bp deletion in exon 2 (*Figure 4—figure supplement 2B*) that is predicted to lead to a premature stop codon at amino acid 82 (*Figure 4—figure supplement 2A*; black box) and exhibit reduced levels of *mfsd2aa* mRNA (*Figure 4—figure supplement 2C*). Homozygous *mfsd2aa* mutants are viable and fertile. While *mfsd2aa* mutants appear in Mendelian ratios at 5 dpf (105/408 fish), a reduced number of *mfsd2aa* mutants survive to adulthood (44/248 fish). *Mfsd2ab*[hm38/hm38] mutants have a 19 bp deletion in exon 5 (*Figure 4—figure supplement 2B*) that is predicted to lead to a premature stop codon at amino acid 175 (*Figure 4—figure supplement 2A*; black box) and exhibit reduced levels of *mfsd2ab* mRNA (*Figure 4—figure supplement 2C*). Homozygous *mfsd2ab* mutants are also viable and fertile and appear in Mendelian ratios both as larvae (51/215 fish) and in adulthood (33/124 fish). Neither mutant displayed obvious angiogenic defects, similar to the normal vasculature observed in mouse *Mfsd2a* mutants (*Ben-Zvi et al., 2014*). In addition to BBB defects, *Mfsd2a* knockout mice also exhibit microcephaly (*Ben-Zvi et al., 2014*). However, neither mutant displayed decreased larval brain size.

To investigate whether either paralogue was necessary for barrier formation, we performed NHS tracer injection assays as we did to identify the developmental timeline of barrier formation and assayed for BBB leakage at 5 dpf. In addition to the use of the exogenous injected fluorescent tracer (1 kDa NHS), we also assayed the leakage of the endogenous transgenic serum DBP-EGFP fusion protein (Tg(*l-fabp:DBP-EGFP*)). *Mfsd2aa* mutants displayed at least a two-fold increase in midbrain and hindbrain parenchymal tracer leakage, both for the injected 1 kDa NHS and the endogenous serum transgene 80 kDa DBP-EGFP, at 5 dpf compared to wild-type sibling controls (*Figure 5*).

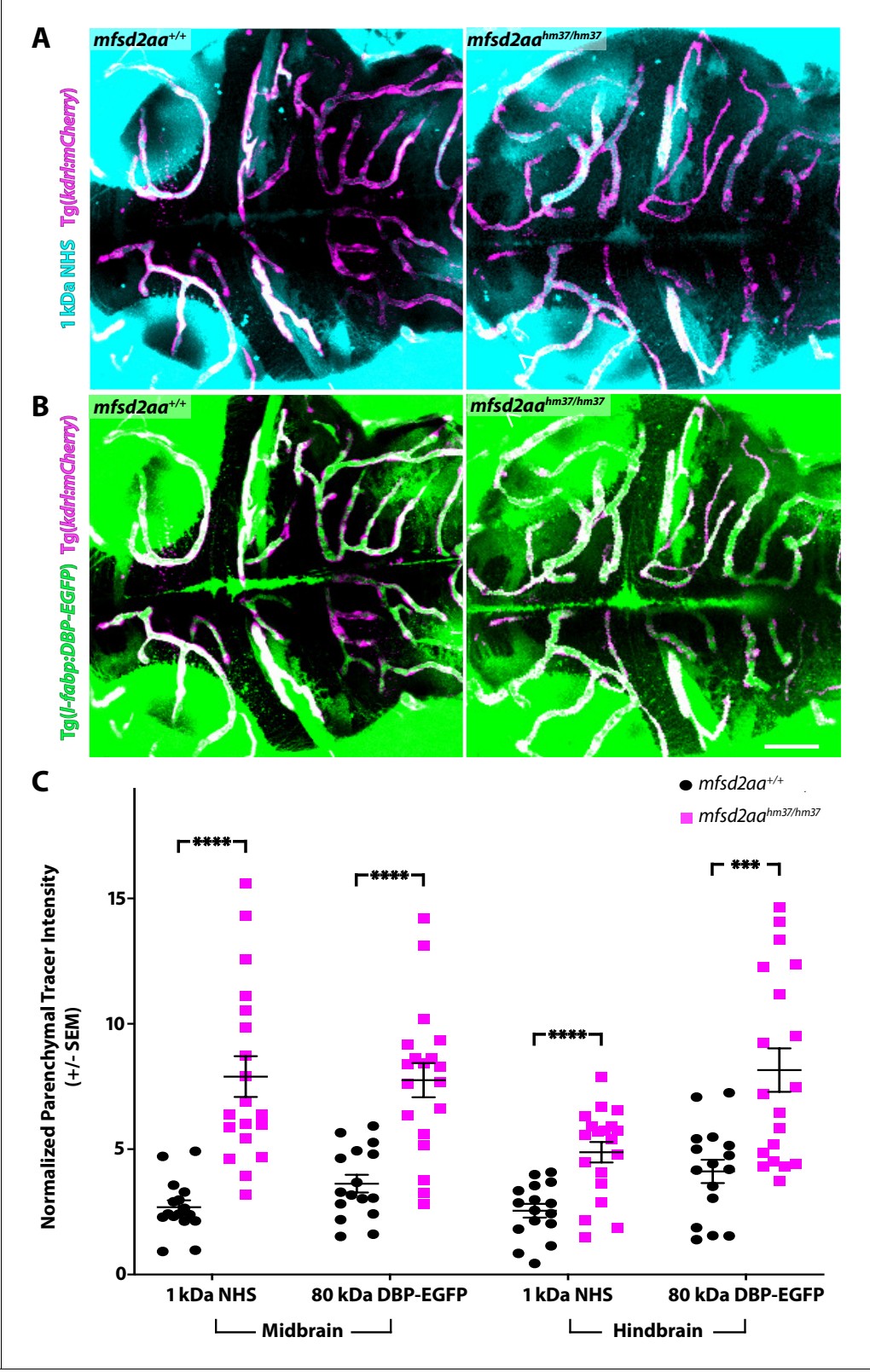

**Figure 5.** *Mfsd2aa* mutants exhibit increased BBB permeability. (**A**) Representative maximum intensity projection images of the brain of wild-type and *mfsd2aa* mutants injected with a fluorescent 1 kDa NHS tracer (turquoise) at 5 dpf. *Mfsd2aa* mutants have increased levels of NHS in the midbrain and hindbrain parenchyma outside of the vasculature (magenta; Tg(*kdrl:mCherry*)) compared to wild-type siblings. (**B**) Representative maximum intensity projection images of the brain of wild-type and *mfsd2aa* mutants expressing the fluorescently labeled 80 kDa transgenic serum protein DBP-EGFP

*Figure 5 continued on next page*

*Figure 5 continued*

(green) at 5 dpf. *Mfsd2aa* mutants have increased levels of DBP-EGFP in the midbrain and hindbrain parenchyma compared to wild-type siblings. The scale bar represents 50 µm. (**C**) Quantification of normalized parenchymal tracer intensity in the midbrain and hindbrain of wild-type (black) and *mfsd2aa* mutants (magenta) reveals that m*fsd2aa* mutants have significantly increased levels of tracer permeability, both for the injected NHS (**A**) and the endogenous transgene DBP-EGFP (**B**). Parenchymal tracer intensity outside of the vasculature was measured and normalized to the blood vessel tracer intensity for both the midbrain and hindbrain in each fish and displayed as a single point. The mean and the standard error are drawn as black lines. ****p<0.0001, ***p<0.001 by t test.

DOI: https://doi.org/10.7554/eLife.47326.017

The following source data and figure supplements are available for figure 5:

**Source data 1.** *Mfsd2aa* mutants exhibit increased BBB permeability, *mfsd2ab* mutants do not.
DOI: https://doi.org/10.7554/eLife.47326.021

**Figure supplement 1.** *Mfsd2ab* mutants do not have altered BBB permeability.
DOI: https://doi.org/10.7554/eLife.47326.018

**Figure supplement 2.** Mosaic *mfsd2aa* crispants display increased BBB permeability while *mfsd2ab* crispants display a wild-type BBB.
DOI: https://doi.org/10.7554/eLife.47326.019

**Figure supplement 3.** *Mfsd2aa*$^{hm37/hm37}$; *mfsd2ab*$^{hm38/hm38}$ double mutants display similar increased BBB permeability to *mfsd2aa*$^{hm37/hm37}$ single mutants.
DOI: https://doi.org/10.7554/eLife.47326.020

These data suggest that mfsd2aa plays a similar role to mouse Mfsd2a in determining barrier properties. Conversely, *mfsd2ab* mutants displayed similarly low levels of parenchymal tracer intensity to wild-type controls at 5 dpf (***Figure 5—figure supplement 1***), indicating that mfsd2ab is dispensable for functional barrier formation. Similarly, when we injected different guides to target either *mfsd2aa* or *mfsd2ab* and assessed BBB permeability in F0 crispants, we observed increased BBB permeability in *mfsd2aa* crispants but not *mfsd2ab* crispants (***Figure 5—figure supplement 2***), confirming the mutant phenotypes. To see if the loss of both paralogues resulted in increased barrier permeability, we investigated *mfsd2aa*$^{hm37/hm37}$; *mfsd2ab*$^{hm38/hm38}$ double mutants for BBB permeability using the endogenous serum DBP-EGFP fusion protein (***Figure 5—figure supplement 3***). While *mfsd2aa* single mutants displayed the previously observed increase in barrier permeability and *mfsd2ab* single mutants did not, *mfsd2aa*$^{hm37/hm37}$; *mfsd2ab*$^{hm38/hm38}$ double mutants displayed similar levels of increased barrier permeability to the *mfsd2aa* single mutants (***Figure 5—figure supplement 3***). These data correspond with previously reported *mfsd2aa* morpholino knockdown leading to increased BBB permeability (***Guemez-Gamboa et al., 2015***). However, neither our *mfsd2aa* or *mfsd2ab* mutants display brain hemorrhage or increased embryonic lethality, as reported in the morphants (***Guemez-Gamboa et al., 2015***). Taken together, these data suggest that the two paralogues, while structurally similar, play different roles, with zebrafish mfsd2aa and mammalian Mfsd2a sharing a conserved role in the BBB.

Since *mfsd2aa* mutants contain a leaky BBB (***Figure 5***), we wanted to further investigate the dynamic nature of the leakage in *mfsd2aa* mutants using time lapse live imaging to determine how tracers enter the mutant brain and whether or not this would mirror the leakage observed in larval fish (***Figure 6***). In 9 *mfsd2aa* heterozygote control fish (***Figure 6—Videos 1*** and ***2***), we observed similarly low rates of Dextran accumulation to that observed in wild-type 5 dpf fish (***Figure 2***). In 9 *mfsd2aa* mutant fish (***Figure 6—Videos 3*** and ***4***), we also never observed any bursting directly from blood vessels as sporadically observed in 3 dpf fish (***Figure 2***). However, we uncovered a significantly increased rate of diffuse Dextran accumulation in the brain parenchyma in *mfsd2aa* mutants (slope of 0.098 intensity/min) compared to heterozygote controls (slope of 0.019 intensity/min; ***Figure 6***). Interestingly, the increased Dextran permeability observed in *mfsd2aa* mutants closely resembles the rate and overall levels of Dextran permeability of the immature 3 dpf larvae (***Figure 2***).

Given the leakage phenotype in *mfsd2aa* mutants at 5 dpf, we next wanted to examine whether the leakage phenotype persisted into adulthood. To address this, we assessed the localization of the serum DBP-EGFP fusion protein in adult brains sections of *mfsd2aa* mutant and wild-type controls. As expected, the wild-type siblings retained the DBP-EGFP within their blood vessels throughout the brain (***Figure 7***). However, *mfsd2aa* mutants exhibited DBP-EGFP extravasation into the brain parenchyma (***Figure 7***), suggesting that the leakage phenotype was not limited to larval fish. Finally, to determine whether this increased permeability was due to increased transcytosis as in *Mfsd2a*

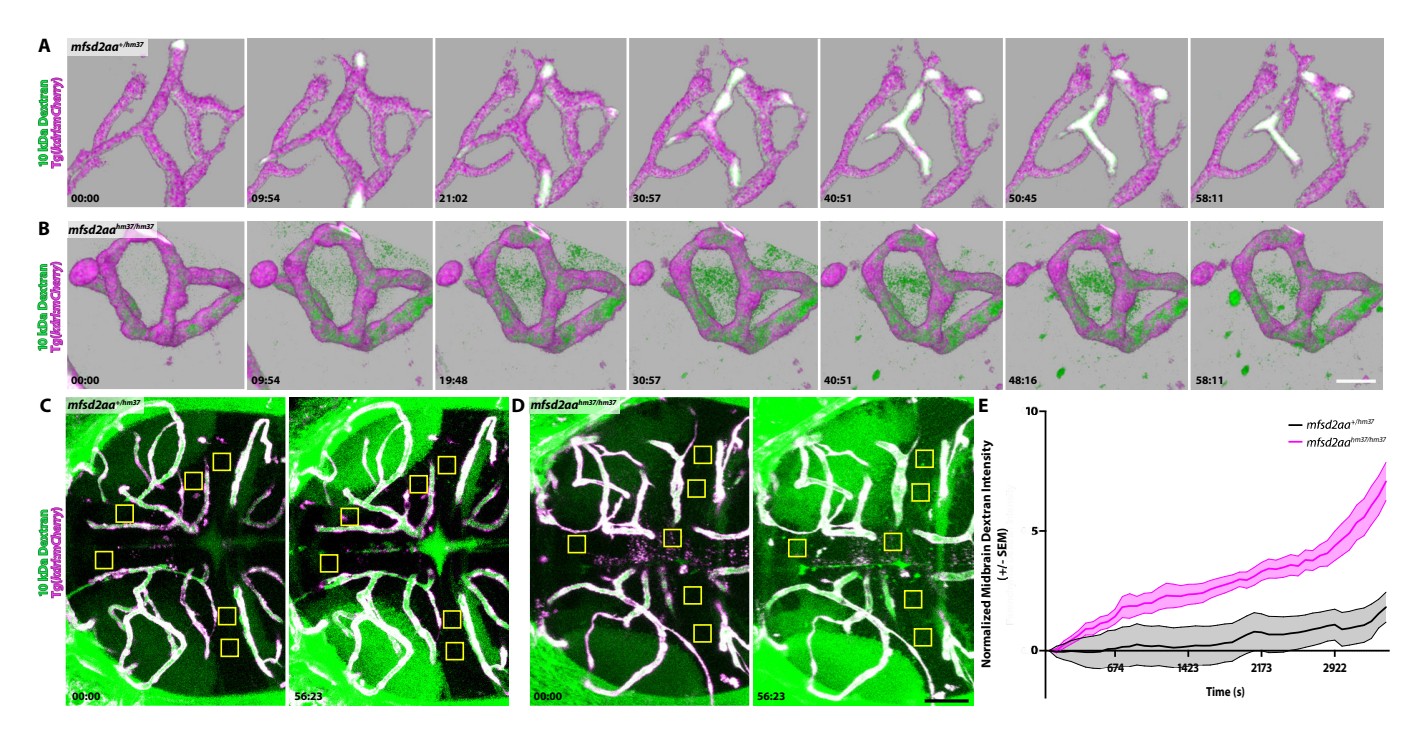

**Figure 6.** Dynamic tracer leakage in *mfsd2aa* mutants via live imaging. (**A**) Time course stills of Dextran tracer dynamics at 5 dpf in *mfsd2aa* heterozygote controls reveals a sealed BBB. (**B**) Time course stills of Dextran tracer dynamics in *mfsd2aa* mutants at 5 dpf reveals a leaky BBB, with increased overall Dextran extravasation into the brain parenchyma. The scale bar represents 20 μm. (**C and D**) Representative dorsal maximum intensity projection of the midbrain of *mfsd2aa* heterozygotes (**C**) and *mfsd2aa* mutants (**D**) at the first and last time point examined. While the heterozygotes restrict the Dextran within the cerebral blood vessels at 5 dpf, *mfsd2aa* mutants exhibit a large increase in overall parenchymal Dextran intensity over the course of 1 hr. Boxed regions are representative of the 6 areas per fish used for analysis in E. The scale bar represents 50 μm. (**E**) Quantification of Dextran intensity in the brain parenchyma over time in heterozygote controls (black) and *mfsd2aa* mutants (magenta) shows a significant difference in tracer leakage dynamics (p<0.0001, Mann Whitney U test), with both more total Dextran accumulation and a faster rate of Dextran accumulation in the mutant brain parenchyma than heterozygote controls. N = 9 fish with 6 regions analyzed and averaged per fish and normalized to Dextran intensity in circulation.

DOI: https://doi.org/10.7554/eLife.47326.022

The following video and source data are available for figure 6:

**Source data 1.** Dynamic tracer leakage in *mfsd2aa* mutants via live imaging.
DOI: https://doi.org/10.7554/eLife.47326.023

**Figure 6—video 1.** Time lapse imaging reveals Dextran impermeability in *mfsd2aa* heterozygote control fish at 5 dpf.
DOI: https://doi.org/10.7554/eLife.47326.024

**Figure 6—video 2.** Time lapse imaging reveals Dextran impermeability in *mfsd2aa* heterozygote control fish at 5 dpf.
DOI: https://doi.org/10.7554/eLife.47326.025

**Figure 6—video 3.** Time lapse imaging reveals increased Dextran permeability in *mfsd2aa* mutant fish at 5 dpf.
DOI: https://doi.org/10.7554/eLife.47326.026

**Figure 6—video 4.** Time lapse imaging reveals increased Dextran permeability in *mfsd2aa* mutant fish at 5 dpf.
DOI: https://doi.org/10.7554/eLife.47326.027

knockout mice, we measured vesicular density in capillaries with luminal diameters less than 5 μm from adult mutant and wild-type siblings using TEM. The *mfsd2aa* mutant blood vessels appeared morphologically normal by TEM, composed of a thin single layer of endothelial cells in close contact with pericytes, as observed in their wild-type siblings (*Figure 8A and B*). A closer examination of the endothelial cells revealed electron-dense tight junction complexes between all apposed endothelial cells in both wild-type and *mfsd2aa* mutant fish (*Figure 8C and D*). However, while wild-type fish display similarly low levels of luminal (0.1 vesicles/μm) and abluminal (0.11 vesicles/μm) vesicular densities to those observed at 7 dpf (*Figure 8C–F*), *mfsd2aa* mutant fish display a significant increase in

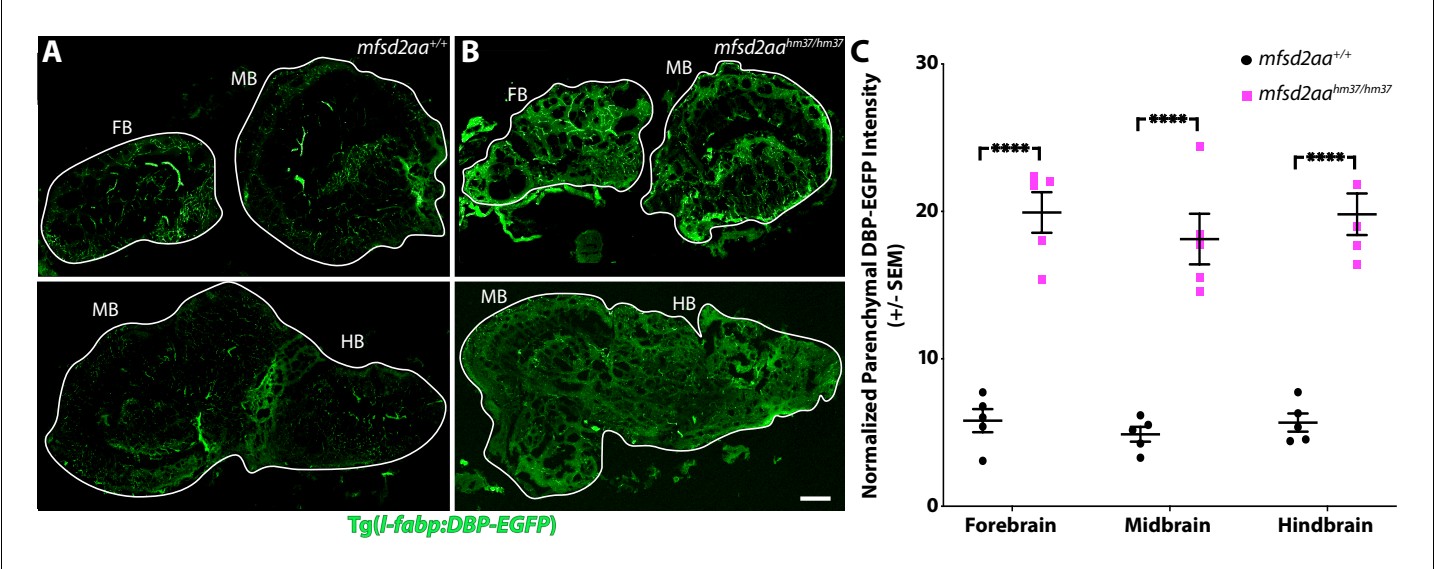

**Figure 7.** Adult *mfsd2aa* mutants display increased BBB permeability. (**A and B**) Sagittal sections of the adult brain of wild-type (**A**) and *mfsd2aa* mutants (**B**) reveals increased extravasation of the transgenic DBP-EGFP serum fusion protein (green) in the forebrain (FB), midbrain (MB) and hindbrain (HB) of *mfsd2aa* mutants compared to wild-type siblings. The scale bar represents 100 µm. (**C**) Quantification of normalized parenchymal DBP-EGFP tracer intensity in the forebrain, midbrain and hindbrain of wild-type (black) and *mfsd2aa* mutants (magenta) reveals that m*fsd2aa* mutants have significantly increased levels of DBP-EGFP permeability throughout the adult brain. Parenchymal tracer intensity outside of the vasculature was measured in 6 distinct areas, averaged and normalized to the blood vessel DBP-EGFP intensity for each region (FB, MB and HB) and displayed as a single point for each fish. The mean and the standard error are drawn as black lines. ****p<0.0001 by t test.

DOI: https://doi.org/10.7554/eLife.47326.028

The following source data is available for figure 7:

**Source data 1.** Adult *mfsd2aa* mutants display increased BBB permeability.

DOI: https://doi.org/10.7554/eLife.47326.029

luminal and abluminal vesicular densities (0.29 and 0.27 vesicles/µm, respectively) compared to wild-type siblings (*Figure 8C–F*). Interestingly this increase in vesicular abundance is even higher than that observed during early barrier development at 3 dpf (*Figure 3*). These data also suggest that the increased tracer leakage observed in *mfsd2aa* mutants results from an increase in vesicular trafficking across the BBB, further supporting a conserved role for mfsd2aa in determining barrier properties. In contrast to *mfsd2aa* mutants, *mfsd2ab* mutants displayed similar levels of abluminal and luminal vesicular pit density to wild-type siblings (*Figure 8—figure supplement 1*), further demonstrating that the paralogue *mfsd2ab* does not play a conserved role in determining barrier properties in zebrafish.

## Discussion

One of the major advantages to studying the BBB in zebrafish is the ability to perform live imaging of tracer permeability dynamics. We provide some of the first data on the dynamics of immature barrier leakage during early development and of barrier leakage seen in a genetic mutant fish at the time when the BBB should be functionally mature. At 3 dpf, we observed two types of leakage, the gradual overall increase in the parenchyma and rare transient bursts, both from a migrating sprout and established blood vessels. We attribute some of the gradual diffuse increase in parenchymal tracer uptake to the high levels of vesicular trafficking observed in our TEM analyses in which the endothelial basement membrane became filled with gold nanoparticles. Time lapse analysis of *mfsd2aa* mutants revealed similar diffuse tracer leakage levels and rates to that observed in the leaky 3 dpf larvae. Since *mfsd2aa* mutant fish have normal angiogenesis but elevated endothelial transcytosis levels, the Dextran leakage at 5 dpf, when their siblings have a functional BBB, provides a direct measure of the in vivo rates and mode of BBB leakage due to unsuppressed transcytosis. The rarer

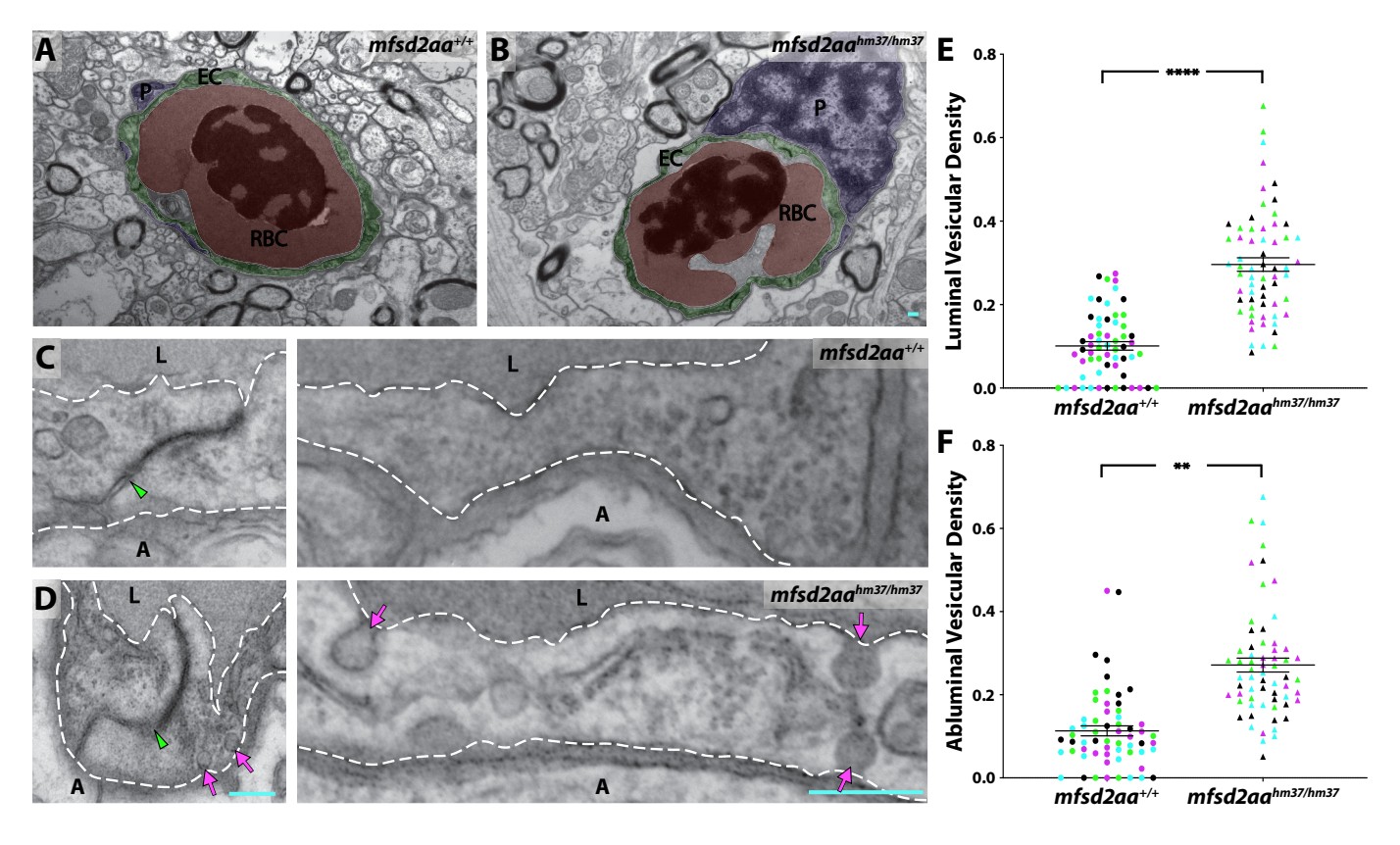

**Figure 8.** M*fsd2aa* mutants exhibit increased transcytosis. (A and B) TEM images of individual blood vessel cross-sections of adult wild-type (A) and *mfsd2aa* mutant fish (B). Endothelial cells (EC) are pseudo-colored green, pericytes (P) are pseudo-colored purple and red blood cells (RBC) are pseudo-colored red. (C and D) High-magnification images of endothelial cells outlined with white dashed lines of wild-type (C) and *mfsd2aa* mutants (D). The images are oriented with the lumen (L) on top and the ablumen (A) on the bottom. *Mfsd2aa* mutants appear to have normal tight junctions (green arrowhead) but elevated levels of luminal and abluminal vesicles (magenta arrows). The scale bars represent 200 nm. (E and F) Quantification of the vesicular densities both on the luminal (E) and abluminal (F) side of the endothelial cells reveals that *mfsd2aa* mutants have increased vesicular densities. N = 4 fish, each marked with a different color, with 15 blood vessels quantified for each fish and displayed as single points. ****p<0.0001, **p<0.01 by nested t test.

DOI: https://doi.org/10.7554/eLife.47326.030

The following source data and figure supplement are available for figure 8:

**Source data 1.** Quantification of adult vesicular densities for *mfsd2aa* and *mfsd2ab* mutants.
DOI: https://doi.org/10.7554/eLife.47326.032

**Figure supplement 1.** M*fsd2ab* mutants exhibit normal vascular maturation.
DOI: https://doi.org/10.7554/eLife.47326.031

bursts of leakage, on the other hand, could be due to fleeting tight junctional ruptures between neighboring endothelial cells rather than increased vesicular trafficking. As these events were extremely scarce during our time lapse imaging, it would be nearly impossible to capture these potential junctional breaches by TEM to confirm this hypothesis. However, with continuing advances in the resolution of fluorescence microscopy, we could use zebrafish to resolve whether these two types of leakage, gradual and bursting, proceed through the same or different subcellular routes into the brain parenchyma, either paracellularly or transcellularly.

While the BBB constitutes the largest interface for blood-brain exchange in adults (*Nag and Begley, 2005*), and represents the most well-studied cellular barrier, it is not the only brain barrier. The blood-cerebrospinal fluid (CSF) barrier is created by the polarized epithelial cells of the choroid plexus (CP) that projects directly into the brain ventricle and creates a barrier between the fenestrated capillaries of the CP and the CSF (*Wolburg and Paulus, 2010*). During development, these

CP epithelial cells become selectively less permissive by expressing tight junction proteins, including Claudin-5 and Occludin, and nutrient transporters, including Mrp1, Mdr1 and Oat3 (*Henson et al., 2014*; *van Leeuwen et al., 2018*). In zebrafish, this blood-CSF barrier has been shown to become completely impermeable by 4 dpf to various molecular weight tracers including 3 kDa fluorescein, 10 kDa and 40 kDa Dextrans (*Henson et al., 2014*). However, while CP epithelial cells are initially permeable to 3 kDa fluorescein and 10 kDa Dextran at 2 dpf, they never allow the passage of the larger 40 kDa Dextran (*Henson et al., 2014*). Since we saw comparable BBB permeability for all tracers used during the developmental time course, including the 80 kDa DBP-EGFP serum transgene, this strongly suggests that the tracers entered the brain predominantly through leaky BBB vessels rather than through an immature blood-CSF barrier. The avascular arachnoid epithelium directly beneath the dura that completely ensheaths the CNS acts as a third barrier between the extracellular fluids of the CNS and the rest of the body (*Abbott et al., 2006*). Previous HRP tracer injections have shown that the zebrafish arachnoid barrier matures after 9 dpf (*Jeong et al., 2008*). Since we did not observe any differences in tracer impermeability between 5 and 10 dpf, when the arachnoid barrier is still leaky, this suggests that the immature arachnoid barrier does not significantly contribute to the observed tracer extravasation into the brain earlier in development. Finally, the similarities between the observed amounts and dynamics of leakage in 3 dpf immature brains and *mfsd2aa* mutant brains at 5 dpf, which contain BBB-specific endothelial transcytosis defects, further suggests that the observed tracer leakage at 3 dpf is predominantly arising through the BBB. Taken together, while alternative routes for tracer entry into the brain exist, including an immature blood-CSF or arachnoid barrier, our methods allowed us to specifically evaluate BBB functionality, and other routes are not significantly contributing to the tracer leakage we observed in the larval brain.

Among the various clathrin-independent transcytotic pathways (*Tuma and Hubbard, 2003*; *Sandvig et al., 2018*), caveolae are particularly abundant in vascular endothelial cells (*Frank et al., 2003*). Caveolae are caveolin-coated 50–100 nm flask-shaped invaginations of the plasma membrane (*Palade, 1953*; *Palade, 1961*). Furthermore, the suppression of caveolae-mediated transcytosis regulates the development and function of both the mouse blood-retinal barrier and the BBB (*Andreone et al., 2017*; *Chow and Gu, 2017*). Taken together, this suggests that the increased transcytosis during early larval stages is most likely caveolae-mediated. Interestingly, the linear profile of Dextran uptake in the midbrain at 3 dpf closely resembles the caveolae-mediated uptake of a fluorescently conjugated aminopeptidase P (APP) antibody in the mouse lung (*Oh et al., 2007*). While similarly linear, the scale is on the order of minutes in the larval zebrafish brain versus seconds in the mouse lung. This discrepancy in timing is most likely due to the large difference in the vesicular densities between the immature BBB endothelium (average of 2 vesicles/$\mu m^2$) and the continuous endothelium of peripheral tissues, which ranges from 30 to 98 vesicles/$\mu m^2$ on the luminal membrane of diaphragm and myocardial endothelium (*Simionescu et al., 1974*). Since loss of caveolin 1 (*Cav1*) is sufficient to precociously seal the mouse blood-retinal barrier at the angiogenic front (*Chow and Gu, 2017*), it would be interesting in future work to examine whether zebrafish *cav1* mutants (*Cao et al., 2016*), which are viable as homozygotes, also exhibit an earlier onset of BBB maturation.

BBB permeability is tightly regulated by endothelial cell interactions with pericytes and astroglial cells. For the first time, we visualized their locations under TEM in zebrafish throughout barrier development. Pericytes have been shown to be essential for establishing the mammalian BBB (*Armulik et al., 2010*; *Bell et al., 2010*; *Daneman et al., 2010*). Similarly, pericyte-deficient notch3[fh332] fish display increased BBB permeability in a tight junction independent manner like pericyte-deficient mice (*Wang et al., 2014*). Our TEM data show that zebrafish pericytes are in close contact with brain endothelial cells (*Figures 2* and *5*), even at the earliest stage examined (3 dpf; *Figure 2*), and are embedded within the endothelial basement membrane as in mammals. Our subcellular localization data is in line with a growing body of evidence for the conserved role and molecular profile of pericytes in the zebrafish BBB (*Wang et al., 2014*; *Ando et al., 2016*; *Lei et al., 2017*; *Vanlandewijck et al., 2018*; *Ando et al., 2019*).

Taken together this study provides a thorough characterization of the development of the zebrafish BBB, highlighting regional differences in timing of functional maturation and capturing the dynamics of the immature BBB. Furthermore, our developmental TEM series demonstrates the role of vesicular trafficking in regulating zebrafish BBB function during development. We have also shown that this down-regulation of vesicular trafficking is necessary for functional BBB formation, as

*mfsd2aa* mutants display increased barrier permeability due to unsuppressed transcytosis. Finally, our time lapse imaging of *mfsd2aa* mutants provides the first direct measure of tracer permeability in vivo that results from increased transcytosis. We hope that this work will serve as a launching pad for future studies using zebrafish to understand the molecular regulators of BBB development and homeostasis in vertebrates.

## Materials and methods

### Zebrafish strains and maintenance

Zebrafish were maintained at 28.5°C following standard protocols (*Westerfield, 1993*). All zebrafish work was approved by the Harvard Medical Area Standing Committee on Animals under protocol number 04487. Adult fish were maintained on a standard light-dark cycle from 9 am to 11 pm. Adult fish, age 3 months to 2 years, were crossed to produce embryos and larvae. For imaging mutant larvae, 0.003% phenylthiourea (PTU) was used beginning at 1 dpf to inhibit melanin production. These studies used the AB and Casper (*mitfa*[w2/w2]; *mpv17*[a9/a9]) (*White et al., 2008*) wild-type strains and the transgenic reporter strains Tg(*l-fabp:DBP-EGFP*)[lri500] (*Xie et al., 2010*), Tg(*kdrl:HRAS-mCherry*)[s896] (*Chi et al., 2008*), abbreviated as Tg(*kdrl:mCherry*) in the text, Tg(*actb2:MA-Citrine*)[hm26] (*Mosaliganti et al., 2012*) and Tg(*actb2:Hsa.H2B-Scarlet*)[hm38].

### Tracer injections

Larvae were immobilized with tricaine and placed in an agarose injection mold with their hearts facing upwards. 2.3 nl of Alexa Fluor 405 NHS Ester (Thermo Fisher: A30000) or Alexa Fluor 647 10 kDa Dextran (Thermo Fisher: D22914) fluorescently conjugated tracers (10 mg/ml) were injected into the cardiac sac using Nanoject II (Drummond Scientific, Broomall, PA). Larvae were then mounted with 1.5% low gelling agarose (Sigma: A9414) in embryo water on 0.17 mm coverslips and imaged live within 2 hr post-injection. For developmental electron microscopy experiments, 2.3 nl of 5 nm NHS-activated gold nanoparticles (Cytodiagnostics: CGN5K–5–1, ~1.1$^{14}$ particles/ml in PBS) were injected into the cardiac sac just as for the fluorescently conjugated tracers. After 5 min of circulation, the fish were fixed for electron microscopy. The brains of transgenic DBP-EGFP adults were fixed in 4% paraformaldehyde in PBS overnight at 4°C. Following fixation the brains were washed three times and sagittally sectioned with a cryostat (30 μm) to reveal DBP-EGFP localization within the brain.

### Transmission electron microscopy (TEM)

Fish were anesthetized with tricaine and initially fixed by immersion in 4% paraformaldehyde (VWR:15713 s)/0.1M sodium-cacodylate (VWR:11653). Following this initial fixation, the larval fish and adults with exposed brains were further fixed for 7–14 days in 2% glutaraldehyde (Electron Microscopy Sciences: 16320)/4% paraformaldehyde/0.1M sodium-cacodylate at room temperature. Following fixation, larvae or dissected brains were washed overnight in 0.1M sodium-cacodylate. Entire larval heads or coronal vibratome free-floating sections of adult brains (50 μm) were post-fixed in 1% osmium tetroxide and 1.5% potassium ferrocyanide, dehydrated, and embedded in epoxy resin. Ultrathin sections of 80 nm were then cut from the block surface and collected on copper grids. The adult sections were counter-stained with Reynold's lead citrate prior to imaging.

### CRISPR mutants

*Mfsd2aa* mutant fish were generated by injection of Cas9 RNA and a single guide RNA (5'-GGTGTG TTTTGCGATCGGAG-3') targeting exon 3 into single-cell fertilized AB wild-type embryos. *Mfsd2ab* mutant fish were generated by injection of Cas9 protein and a single guide RNA (5'-TGAGAGCA-GAGTAGGGCACG-3') targeting exon 5 into single-cell fertilized double transgenic Tg(*l-fabp:DBP-EGFP; kdrl:mCherry*) embryos. F0 injected fish were raised, outcrossed to wild-type fish and screened for potential mutant founders by PCR and sequencing. Founders were outcrossed to wild-type fish at least three times before leakage analyses. Heterozygous fish were intercrossed for all leakage assays, with the exception of the *mfsd2aa* time lapse microscopy, where a heterozygous fish was crossed to a homozygous mutant. All larvae were imaged prior to genotyping to identify wild-type and mutant fish. The stable *mfsd2aa* mutant line was genotyped using 5'-AAATCACCTC

TTCCAGTGAGGA-3' and 5'-ATAGTAACAAAACGATGCTGAGCC-3' primers followed by a Pvu1 restriction digest, which does not digest the mutant DNA. *Mfsd2ab* mutants were genotyped using 5'-GTCTACTCCATTTGCTGTACTTTGC-3' and 5'-AGAGCAGAGTAGGGCACGTGGAAGC-3' primers for the wild-type allele and 5'-GTCTACTCCATTTGCTGTACTTTGC-3' and 5'-GTGCTGATGAAGCAC TGATGAAGCA-3' for the mutant allele.

*Mfsd2aa* crispant fish were generated by injection of Cas9 protein and two new guide RNAs (5'-GCTCACTAAAAACCCAACGG-3' and 5'-TTGTAGAAAGAAGCCCAGGG-3') targeting exons 2 and 3 into single-cell fertilized double transgenic Tg(*l-fabp:DBP-EGFP; kdrl:mCherry*) embryos. Similarly, *mfsd2ab* crispants were generated by injection of Cas9 protein and two new guide RNAs (5'-AAGC-GAGATGGCAAAAGGAG-3' and 5'- ACCCACAAACAAGATGATGG-3') targeting exons 1 and 3 into single-cell fertilized double transgenic Tg(*l-fabp:DBP-EGFP; kdrl:mCherry*) embryos. F0 crispants were analyzed for leakage at 5 dpf.

## qPCR

qPCR was performed on cDNA obtained from wild-type, *mfsd2aa*[hm37], *mfsd2ab*[hm38], *mfsd2aa* and *mfsd2ab* F0 crispant 5 dpf whole larvae mRNA. An Eppendorf Realplex2 Real-Time PCR system was used for qPCR experiments and gene expression levels were normalized relative to that of a reference gene, *18*s using primers 5'-TCGCTAGTTGGCATCGTTTATG-3' and 5'-CGGAGGTTCGAA-GACGATCA-3'. Levels of *mfsd2aa* were measured using 5'-CTCTTCACTTCGCTAGCCTTCATG-3' and 5'-CGATGTAAACAGCAGTCTTTTTCCC-3' primers and *mfsd2ab* levels were measured with 5'-TCTCGACTCTTAGTCTTGATTTCGC3' and 5'-GAGTCCGTTTCTGAATCCATCTCG-3'. All reactions were performed in technical duplicates, and the results represent four independent biological samples including the SEM.

## HCR fluorescent in situ hybridization (FISH)

HCR RNA in situ hybridization (Molecular Instruments) experiments on 14 µm cryosections of fixed 3 and 5 dpf larvae and adult brains were performed as previously described (*Choi et al., 2018*). Briefly, sections were air dried and re-fixed in 4% paraformaldehyde in PBS for 10 min at room temperature. Following fixation, sections were further permeabilized using 1 µg/ml Proteinase K (ThermoFisher) for 5 min, followed by PBS washes. Slides were then put through an ethanol dehydration series of 50%, 70% and two rounds of 100% ethanol and air dried for 5 min. Dried slides were put into pre-hyb solution for at least 10 min at 37°C. Subsequently, the probes for *mfsd2aa, mfsd2ab, slc2a1a* and mouse *Gfap* were added to the slides at a final concentration of 4 nM to hybridize overnight at 37°C. The next day, slides were washed with a series of wash buffer to 5x SSCT (5x SSC with 0.1% Tween 20). Excess liquid was then removed and samples were immersed in amplification buffer at room temperature for 30 min prior to hairpin amplification, which occurred overnight at room temperature. The next day, slides were washed in 5x SSCT and mounted with Fluoromount-G (Electron Microscopy Sciences).

## Imaging

All imaging of tracer permeability was performed on a Leica SP8 laser scanning confocal microscope using the same acquisition settings using a 25x water immersion objective. Time lapse imaging was performed on the SP8 using a resonance scanner. FISH sections were also imaged with the Leica SP8 using a 63x oil immersion objective. A 1200EX electron microscope (JOEL) equipped with a 2 k CCD digital camera (AMT) was used for all TEM studies. Images were visualized and quantified using ImageJ (NIH) and Adobe Photoshop. Time lapse videos were visualized as 3D reconstructions and cropped to highlight particular blood vessels of interest using Imaris software (Bitplane).

## Tracer permeability quantification

All quantifications were performed on blinded image sets. For static images, parenchymal fluorescent tracer intensity was measured using ImageJ in the entire regional parenchyma outside of the vasculature in 30 µm thick maximum intensity projections of the larval brains. These projections began on average 15 µm below the mesencephalic vein to reduce the effects of potential leakage diffusion from the surface vessels and had the vasculature masked and removed for intensity quantification. These parenchymal tracer intensity values were then normalized to the tracer intensity

within the vasculature to account for differential amounts of circulating tracer. For time lapse imaging, Dextran intensity was measured in 6 parenchymal regions of average intensity projections of the time lapse videos and averaged as a single value per fish. These values were normalized to the average fluorescence intensity in the lumen at each time point. Z-stacks were collected with 1 μm z-steps using a 25x water immersion objective on a Leica SP8 laser scanning confocal microscope.

### TEM quantification

For all TEM quantifications, vesicular density values were calculated from the number of non-clathrin coated flask-shaped vesicles per μm of endothelial luminal or abluminal membrane for each image collected. Larval basement membrane gold intensity was measured in 3 regions per endothelial cell and normalized to luminal gold intensity values within individual images using ImageJ. All images for analysis were collected at 12,000 x magnification on the JOEL 1200EX electron microscope. 10–15 vessels were quantified for each fish, with each color representing a different fish.

### FISH quantification

For all FISH experiments, we quantified the number of gene expression puncta within the vasculature of maximum intensity projections using ImageJ. These puncta were normalized to total blood vessel area within the image. Non-specific background levels of mouse *Gfap probe* binding were similarly assessed at all three time points. These background levels were then subtracted from *mfsd2aa* and *mfsd2ab* expression levels and negative values were set to 0.

### Statistical analysis

All statistical analyses were performed using Prism 8 (GraphPad Software). Two group comparisons were analyzed using an unpaired two-tailed t test. Nested t tests were employed for all electron microscopy comparisons to account for the actual N versus vessels analyzed. Multiple group comparisons were analyzed with two-way ANOVA, followed by a post hoc Tukey's multiple comparison test. The time lapse leakage dynamics were analyzed with a Mann-Whitney U test to discriminate whether the two patterns of leakage accumulation were different. Sample size for all experiments was determined empirically using standards generally employed by the field, and no data was excluded when performing statistical analysis. Standard error of the mean was calculated for all experiments and displayed as errors bars in graphs. Statistical details for specific experiments, including exact n values and what n represents, precision measures, statistical tests used, and definitions of significance can be found in the Figure Legends.

## Acknowledgements

We thank members of the Gu and Megason laboratories for data discussion and comments on the manuscript; Dr. Zach O'Brown for discussions and comments on the manuscript; Dr. Bela Anand-Apte (Cleveland Clinic) for providing the transgenic Tg(*l-fabp:DBP-EGFP)* fish line (*Xie et al., 2010*) and Dr. Leonard Zon for providing the transgenic Tg(*kdrl:HRAS-mCherry*) line; and the HMS Electron Microscopy Core Facility, with special thanks to Louise Trakimas for all of her assistance in troubleshooting and preparing the TEM samples. This work was supported by the Damon Runyon Cancer Research Foundation (NMO), the Mahoney Postdoctoral Fellowship (NMO), the Fidelity Biosciences Research Initiative (CG), and the National Institutes of Health Pioneer Award DP1 NS092473 (CG). The research of CG was also supported in part by a Faculty Scholar grant from the Howard Hughes Medical Institute.

## Additional information

### Funding

| Funder | Grant reference number | Author |
| --- | --- | --- |
| Damon Runyon Cancer Research Foundation | Postdoctoral Research Fellowship | Natasha M O'Brown |
| Fidelity Biosciences | | Chenghua Gu |

| National Institutes of Health | National Institutes of Health Director's Pioneer Award DP1 NS092473 | Chenghua Gu |
| Harvard Medical School | Mahoney Postdoctoral Fellowship | Natasha M O'Brown |

The funders had no role in study design, data collection and interpretation, or the decision to submit the work for publication.

## Author contributions
Natasha M O'Brown, Conceptualization, Data curation, Formal analysis, Funding acquisition, Investigation, Visualization, Methodology, Writing—original draft, Project administration, Writing—review and editing; Sean G Megason, Conceptualization, Supervision, Methodology, Writing—original draft, Writing—review and editing; Chenghua Gu, Conceptualization, Supervision, Funding acquisition, Writing—original draft, Project administration, Writing—review and editing

## Author ORCIDs
Natasha M O'Brown (iD) https://orcid.org/0000-0002-0718-7037
Sean G Megason (iD) https://orcid.org/0000-0002-9330-2934
Chenghua Gu (iD) https://orcid.org/0000-0002-4212-7232

## Ethics
Animal experimentation: This study was performed in strict accordance with the recommendations in the Guide for the Care and Use of Laboratory Animals of the NIH. All animals were handled according to approved institutional animal care and use committee (IACUC) protocols (protocol 04487) of Harvard Medical School. All work was approved by the Harvard Medical Area Standing Committee on Animals.

## Decision letter and Author response
Decision letter https://doi.org/10.7554/eLife.47326.035
Author response https://doi.org/10.7554/eLife.47326.036

# Additional files

## Supplementary files
• Transparent reporting form
DOI: https://doi.org/10.7554/eLife.47326.033

## Data availability
All raw data are attached as an Excel spreadsheet.

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
