## [Decision Letter]

Thank you for submitting your article "Suppression of transcytosis regulates zebrafish blood-brain barrier development" for consideration by *eLife*. Your article has been reviewed by three peer reviewers, and the evaluation has been overseen by a guest Reviewing Editor and Didier Stainier as the Senior Editor. The following individuals involved in review of your submission have agreed to reveal their identity: Benoit Vanhollebeke (Reviewer #1); Michael R Taylor (Reviewer #2); Kelly Monk (Reviewer #3).

The reviewers have discussed the reviews with one another and the Reviewing Editor has drafted this decision to help you prepare a revised submission.

All of the reviewers thought that the work was of potential interest, but that there were a number of major concerns that need to be addressed.

Based on the discussion the reviewers specific points included that: "there were several concerns from all reviewers regarding interpretation of data and terminology that should be addressed both in the major and minor comments" and "a more careful job with the analysis of the new mutants they have generated is required." Furthermore, all reviewers agreed that counting cells with tracer as a proxy for BBB leakage was concerning as this could be a neural intrinsic feature and BBB-independent process, and that the dynamic range of this assay remains unclear as changing the molar amounts of tracer does not lead to changes in cell counts.

The authors should address each of the concerns raised by the reviewers.

*Reviewer #1:*

This study by Brown et al. investigates BBB maturation in zebrafish larvae combining time-lapse confocal imaging of tracer extravasation, genetics and electron microscopy. By using different tracers, the authors assess the time-point at which developmental BBB maturity is reached in a spatiotemporal fashion. The decrease in leakage in the midbrain at 5 days post fertilization was further explored by TEM and shown to correlate with a decrease in intracellular transcytosis. The sealing of the blood brain barrier, correlating with a decrease in transcytosis, was further explored by studying the conserved function of Mfsd2a in this process.

The paper is of interest as it further documents the similarities between the zebrafish and the mammalian BBB structures. The TEM studies, combined with *mfsd2aa* mutants convincingly show that transcytosis suppression contributes to BBB maturation in zebrafish, as it does in the mouse.

The authors convincingly report two apparently independent tracer accumulation modes in the zebrafish larval brain. The first one, previously observed by other investigators, results in diffuse tracer accumulation across the parenchyma, follows near-linear kinetics and, in the context of this study, has been analyzed in the midbrain after at 3 and 5 dpf. The other mode is interesting as so far unreported and results in focal tracer accumulation secondary to perivascular "bursts" that can be captured by time-lapse confocal microscopy.

The authors propose that the steady and diffuse accumulation is secondary to transcytosis across the immature BBB endothelium and is controlled developmentally in the midbrain by *mfsd2aa*, while the other would result from sporadic vessel instability and uptake through what are referred to as "parenchymal cells". The number of such tracer-filled cells is then quantified as a proxy for tracer leakage across the BBB to infer both BBB closure and asses genetic mutants.

While overall interesting and timely, some of the leakage assay methodology central to the message of this manuscript might suffer from technical caveats that could be addressed before publication.

Major comments:

First. While the confocal time-lapse microscopy evidence is strong enough to support the idea that the perivascular bursts are secondary to leakage through the BBB endothelium, there is no direct evidence that the more distant focal accumulation, as well as the disuse accumulation throughout the parenchyma, results from leakage through the immature BBB. When dyes are injected in the zebrafish cardiovascular system, they can reach the brain parenchyma through multiple routes, with only one of them being the BBB. Jeong et al., 2008 reported that several mesencephalic vessels (that are not intraneural BBB vessels) are leaky at 3 dpf. These vessels are therefore a likely alternative tracer entry point to the nearby midbrain. Similarly, upon intracardial injection, tracers of any kind accumulate in the CSF within ventricular spaces, and could therefore reach the brain through the ependymal cell layer. The poorly characterized meningeal barriers (arachnoid, pia mater, glia limitans) constitute yet another portal to the brain. Of note, these latter were shown to close at much later stages (after 9 dpf) by Jeong et al., 2008. Therefore, tracer accumulation in the larval zebrafish brain does not necessarily imply leakage through the BBB. Of note, this challenge is particularly acute due the small size of the larval zebrafish brain, that in contrast to the mammalian structures, poorly allows for anatomical leakage inferences. Admittedly, addressing this issue experimentally is challenging and well beyond the scope of what can be achieved at the revision step of this particular study. In this reviewer's opinion, these anatomical considerations should however be carefully discussed by the authors as they impact directly several of the main conclusions of the manuscript.

Overall, we recommend rephrasing the manuscript in order to:

i) shift away from a BBB-centric view and give more space to the anatomical complexity of the multiple brain barriers. We would for instance suggest rephrasing the first section heading from ("P-A gradient of zebrafish BBB development" to "P-A gradient of zebrafish brain barriers development").

ii) Importantly, avoid any firm causality inferences between BBB phenotypes (like the presence or absence of vesicular transport) and at distance tracer accumulation. For instance, in the Abstract the sentence "Electron microscopy studies further reveal that this steady accumulation results from high levels of transcytosis that are eventually suppressed, sealing the BBB" should be rephrased or removed. Of note, this is not to say that the authors shouldn't discuss on the developmental correlation of transcytosis suppression and reduced tracer accumulation in the Discussion section. Similarly, the HRP leakage in the adult *mfsd2aa* mutants are convincing.

Second. While this reviewer agrees that it seems reasonable that the focal tracer accumulation indeed reflects intracellular accumulation in "scavenger " parenchymal cells, the authors fail to present evidence to support this claim. When considering carefully the Videos 1 and 2, tracer extravasation and "cellular" accumulation appear as a synchronous and perhaps unique process, which is puzzling. Moreover, the still images and the video recording these sporadic bursts do not allow discriminating with the alternative scenario of focal accumulation in extracellular spaces, or labelling of apoptotic cells or cell debris. As the authors use these "cells" as a "proxy of tracer leakage", this requires clarification. We suggest approaching this question by confocal imaging with counterstained nuclei and cell membranes.

Third. The authors propose a model in which unrestricted transcytosis triggers diffuse tracer accumulation, a process negatively regulated by *mfsd2aa* and quantified in WT animals in Figure 2E. Surprisingly, in *mfsd2aa* mutants, quantification of this mode of leakage was not performed. This is essential to support the model and should be performed.

Fourth. Instead, "parenchymal cells" were monitored (see comment #2). We were not convinced that monitoring the number of "cells" is predictive to leakage extent. It would only be so if (i) the total number of cells is high, stable during development and unaltered in genetic mutants and if (ii) the amount of substrate tracer is limiting so that the more tracer reaches the parenchyma, the more cells are detectable. None of these conditions seem established in light of the data presented. For instance, as the nature of the presumed cells is unknown, their total number (labeled or not) cannot be established and therefore different time points and genetic conditions cannot be compared (Figure 1D, Figure 1—figure supplement 1, Figure 4C, Figure 4—figure supplement 2, 3). How can the authors rule out that the total number of this presumptive cell type is not merely increased in *mfsd2aa* mutants? According to previous reports *mfsd2aa* is widely expressed throughout the zebrafish embryonic and larval brain. Similarly, how can the authors exclude that the number of these cells drops over time in the midbrain? Instead of counting "cells", the mean intensity of the "cells" could be analyzed (alternatively, asses the parenchymal leak, see third comment). If transcytosis (or more broadly tracer leakage) is increased, the anticipated result is an increase in the average fluorescence intensity of cell-associated label (rather than an increase in the actual number of cells).

Assessing the number of cells as a readout for transcytosis seems moreover also puzzling from cellular standpoints. Indeed, at least some of these "cells" were shown to be filled secondary to sudden rupture of endothelial integrity (Figure 2E) and therefore would not be a faithful readout for transcytosis upregulation. Final sentence in subsection “Posterior-Anterior Gradient of Zebrafish BBB Development” are unclear in that respect as well. We would agree with the authors that tight junctional defects would likely exhibit size-selective leakage properties (although that remains to be shown in zebrafish). Consequently, the observed labelling of the "parenchymal cells" with tracers of all tested sizes suggest an alternative leakage mode is at play (if we assume this is secondary to BBB defects, see above). It does however not discriminate vessel rupture from transcytosis. So in absence of vessel rupture quantification in *mfsd2aa* mutants versus WT, interpretation of the cell counts is ambiguous.

*Reviewer #2:*

1) This reviewer believes that the title of this manuscript overstates the conclusions of the study. To conclude that BBB development is regulated by suppression of transcytosis, implies that other characteristics of BBB development are also controlled by this cellular process. As BBB development requires the acquisition of many properties (e.g. formation of tight junctions, expression of multiple types of transporters, establishment of cellular interactions, etc), including the suppression of transcytosis as described here, please consider a title that more accurately reflects the results of the study.

2) The conclusions of the manuscript rely significantly upon the spatiotemporal expression of *mfsd2aa* (and *mfsd2ab* to a lesser extent). Based upon the results of this study, it would be predicted that *mfsd2aa* is not expressed in brain endothelial cells at 3 dpf, which allows for transcytosis of tracers into the brain parenchyma. Furthermore, it would be predicted that *mfsd2aa* is expressed in brain endothelial cells by 5 dpf, which then suppresses transcytosis of tracers into the brain parenchyma. However, the authors do not provide any expression data for either *mfsd2aa* or *mfsd2ab*.

In previous studies, Guemez-Gamboa et al. used whole-mount in situ hybridization to show that both zebrafish *mfsd2aa* or *mfsd2ab* transcripts are expressed throughout the nervous system in zebrafish at 24, 48, and 96 hpf. However, it is not clear from this data whether brain endothelial cells express either transcript. In addition, Thisse and Thisse showed basal expression of *mfsd2aa* in spinal cord neurons as early as 20-25 somites to Prim-5 stage, no spatial restriction at the Prim-15 to Prim-25 and High-pec to Long-pec stages, and neurocranium expression at 5 dpf (https://zfin.org/ZDB-GENE-041114-166/expression). Thus, it remains to be determined whether zebrafish *mfsd2aa* or *mfsd2ab* is expressed in brain endothelial cells.

Please provide expression data that demonstrates *mfsd2aa* expression (or lack of expression) in zebrafish brain endothelial cells at the appropriate developmental time points (i.e. 3 dpf, 5 dpf, and adult). Perhaps another valid strategy would be to rescue *mfsd2aa* mutants by cell autonomous expression of wild-type *mfsd2aa* in endothelial cells.

Guemez-Gamboa et al. (2015) Nature Genetics 47:809-813

Thisse and Thisse, (2004) ZFIN Direct Data Submission (http://zfin.org).

3) To quantify "leakiness", this study focused on the number of parenchymal cells that accumulated tracer in the midbrain and demonstrated that larvae at 3 dpf are leakier than 5 dpf. The authors also indicated that parenchymal cells in the hindbrain did not accumulate tracer at a significant level at either developmental time point. However, in the *mfsd2aa* mutants, the authors did not examine midbrain leakiness at 3 dpf or in the hindbrain at either 3 dpf or 5 dpf. Do the *mfsd2aa* mutants at 3 dpf exhibit increased leakiness compared to wild-types at 3 dpf? Also, if *mfsd2aa* suppresses transcytosis in all brain endothelial cells, then is the hindbrain leakier in *mfsd2aa* mutants at 3 dpf and 5 dpf? If not, please explain.

*Reviewer #3:*

This manuscript from O'Brown and colleagues represents the most careful analysis of zebrafish blood brain barrier (BBB) development performed to date. Given the genetic tractability of zebrafish, the unparalleled ability to perform in vivo imaging, and the amenability to small molecule screens, zebrafish is an ideal vertebrate model system for BBB study. Previous work raised a controversy as to whether the BBB forms in zebrafish by 3 days of age (Jeong et al., 2008; Umans et al., 2017) or between 3-10 days of age (Fleming et al., 2013). It is critically important to resolve this controversy and to define how the BBB forms in zebrafish to set the stage for future mechanistic, genetic, and drug discovery studies. For example, one can imagine small molecule screens in zebrafish larvae to find compounds that selectively open the BBB for therapeutic drug delivery. Without fully understanding how the BBB forms in zebrafish, however, these studies cannot be performed. O'Brown et al. begin to tackle this important problem by thoroughly characterizing BBB formation in zebrafish between 3-6 days of age. Using tracer injections, live imaging, and electron microscopy, they show that the hindbrain BBB is closed by 3 days, whereas the midbrain closes between 3-5 days. This developmental gradient is similar to mouse. Given that the lipid transporter Mfsd2a is required for proper BBB formation in mammals, O'Brown and colleagues also generate and analyze new mfsd2a zebrafish mutants. Mutant analysis indicates that the function of Mfsd2a is conserved from zebrafish to mammals.

There are some minor experimental additions that could be performed to strengthen the study's conclusions and impact.

1) Given that antibodies to mark tight junctions exist that work in zebrafish, it would be useful to add this analysis to the time course of BBB formation.

2) Given the controversy surrounding the timing of BBB closure in zebrafish, it is important to extend the time course to 10 days to be consistent with the Fleming et al. study. Additionally, the hindbrain at 3 days shows ~2 cells/embryo taking up tracer whereas the authors consider the midbrain BBB closed at 5-6 days with ~8 cells/larva taking up tracer. Thus, it is important to test if the number of cells taking up tracer further reduces with time.

3) The standard in the field is to rescue a new mutant with wild-type gene product to control for off-target CRISPR/Cas9 effects. Similarly, qPCR for *mfsd2aa* and *mfsd2ab* are important controls for the new mutants, especially since phenotypes were not observed in *mfsd2ab* mutants.

4) Not performing in vivo imaging of the *mfsd2aa* mutants seems like a missed opportunity. Such investigation could also reveal if both types of leakage observed in the time course are affected in *mfsd2aa* mutants. The analysis presented in Figure 2E would also be helpful to show for the mutants.

Overall this is an important descriptive study for both zebrafish development and BBB fields that will be foundational for future work in many laboratories.

---

## [Author Response]

The authors should address each of the concerns raised by the reviewers.

Reviewer #1:

[…] Overall, we recommend rephrasing the manuscript in order to:i) shift away from a BBB-centric view and give more space to the anatomical complexity of the multiple brain barriers. We would for instance suggest rephrasing the first section heading from ("P-A gradient of zebrafish BBB development" to "P-A gradient of zebrafish brain barriers development").ii) Importantly, avoid any firm causality inferences between BBB phenotypes (like the presence or absence of vesicular transport) and at distance tracer accumulation. For instance, in the Abstract the sentence "Electron microscopy studies further reveal that this steady accumulation results from high levels of transcytosis that are eventually suppressed, sealing the BBB" should be rephrased or removed. Of note, this is not to say that the authors shouldn't discuss on the developmental correlation of transcytosis suppression and reduced tracer accumulation in the Discussion section. Similarly, the HRP leakage in the adult mfsd2aa mutants are convincing.

We thank the reviewer for his thoughtful remarks. We closely examined the literature about alternative routes into the brain during development and have added a paragraph to the discussion talking about the other brain barriers, their developmental timing and restrictiveness during development, as well as potential contribution to the leakage during early development. Based on this thorough literature review and our cumulative findings, we conclude that Figure 1 is mostly representing BBB leakage as discussed in subsection “Conserved Role of *Mfsd2a* in Determining BBB Function” as well as in the Discussion section. Overall, we have tempered our interpretations of the causality of tracer leakage between different experiments throughout the manuscript. For example, we have altered the referred to sentence in the Abstract to “Electron microscopy studies further reveal high levels of transcytosis in brain endothelium early in development that are suppressed later. The timing of this suppression of transcytosis coincides with the establishment of BBB function.”

Second. While this reviewer agrees that it seems reasonable that the focal tracer accumulation indeed reflects intracellular accumulation in "scavenger " parenchymal cells, the authors fail to present evidence to support this claim. When considering carefully the Videos 1 and 2, tracer extravasation and "cellular" accumulation appear as a synchronous and perhaps unique process, which is puzzling. Moreover, the still images and the video recording these sporadic bursts do not allow discriminating with the alternative scenario of focal accumulation in extracellular spaces, or labelling of apoptotic cells or cell debris. As the authors use these "cells" as a "proxy of tracer leakage", this requires clarification. We suggest approaching this question by confocal imaging with counterstained nuclei and cell membranes.

We agree with this reviewer, and also reviewer 2, that the “scavenger” parenchymal cells should be better characterized. In order to address these concerns, we have performed these same tracer leakage assays in membrane labelled fish and have observed both the diffuse accumulation in the extracellular spaces and the cellular accumulation within cell membranes and have included this data as a supplement (Figure 2—figure supplement 1). In addition, we have also performed these tracer leakage assays in nuclear labelled transgenic fish and have observed co-localization between the injected Dextran and the nuclear signal, further supporting that the injected tracers can be taken up by “scavenger” parenchymal cells (Figure 2—figure supplement 1).

Third. The authors propose a model in which unrestricted transcytosis triggers diffuse tracer accumulation, a process negatively regulated by mfsd2aa and quantified in WT animals in Figure 2E. Surprisingly, in mfsd2aa mutants, quantification of this mode of leakage was not performed. This is essential to support the model and should be performed.

We thank the reviewer for this comment and wholeheartedly agree that this is a crucial experiment to support the notion of increased transcytosis altering BBB leakage. We have performed the suggested time lapse leakage assays in *mfsd2aa* mutant and heterozygote control fish to observe the dynamics of leakage in a system with BBB-specific increased transcytosis. We observed similar diffuse leakage into the brain parenchyma in *mfsd2aa* mutants like we did at 3 dpf (Figure 6), both with similar overall levels and rates of leakage into the brain parenchyma.

Fourth. Instead, "parenchymal cells" were monitored (see comment #2). We were not convinced that monitoring the number of "cells" is predictive to leakage extent. It would only be so if (i) the total number of cells is high, stable during development and unaltered in genetic mutants and if (ii) the amount of substrate tracer is limiting so that the more tracer reaches the parenchyma, the more cells are detectable. None of these conditions seem established in light of the data presented. For instance, as the nature of the presumed cells is unknown, their total number (labeled or not) cannot be established and therefore different time points and genetic conditions cannot be compared (Figure 1D, Figure 1—figure supplement 1, Figure 4C, Figure 4—figure supplement 2, 3). How can the authors rule out that the total number of this presumptive cell type is not merely increased in mfsd2aa mutants? According to previous reports mfsd2aa is widely expressed throughout the zebrafish embryonic and larval brain. Similarly, how can the authors exclude that the number of these cells drops over time in the midbrain? Instead of counting "cells", the mean intensity of the "cells" could be analyzed (alternatively, asses the parenchymal leak, see third comment). If transcytosis (or more broadly tracer leakage) is increased, the anticipated result is an increase in the average fluorescence intensity of cell-associated label (rather than an increase in the actual number of cells).Assessing the number of cells as a readout for transcytosis seems moreover also puzzling from cellular standpoints. Indeed, at least some of these "cells" were shown to be filled secondary to sudden rupture of endothelial integrity (Figure 2E) and therefore would not be a faithful readout for transcytosis upregulation. Final sentence in subsection “Posterior-Anterior Gradient of Zebrafish BBB Development” are unclear in that respect as well. We would agree with the authors that tight junctional defects would likely exhibit size-selective leakage properties (although that remains to be shown in zebrafish). Consequently, the observed labelling of the "parenchymal cells" with tracers of all tested sizes suggest an alternative leakage mode is at play (if we assume this is secondary to BBB defects, see above). It does however not discriminate vessel rupture from transcytosis. So in absence of vessel rupture quantification in mfsd2aa mutants versus WT, interpretation of the cell counts is ambiguous.

We truly thank this reviewer (and reviewer 2) for this comment and we have now analyzed the data in an alternative way as shown in Figure 1 and described in the Materials and methods section. We now measure the overall tracer intensity in the brain parenchyma normalized to tracer intensity within circulation, as we had previously done for the time lapse imaging, to quantify leakage across developmental stages and mutant lines. Interestingly, this analysis revealed high tracer permeability throughout the larval brain at 3 dpf, whereas our previous cell counting method showed the hindbrain barrier had been sealed at this time point based the lack of “scavenger” cells in the hindbrain. This difference could be due to the sensitivity of two quantification methods or the lack of these particular scavenger cells in the hindbrain. However, this has not altered our overall assessment of a posterior to anterior gradient of barrier sealing, as at 4 dpf, we observed a sealed hindbrain barrier but a leaky midbrain barrier and at 5 dpf, everything appeared sealed. The combination of both methods for leakage quantification strengthened this study.

Reviewer #2:

1) This reviewer believes that the title of this manuscript overstates the conclusions of the study. To conclude that BBB development is regulated by suppression of transcytosis, implies that other characteristics of BBB development are also controlled by this cellular process. As BBB development requires the acquisition of many properties (e.g. formation of tight junctions, expression of multiple types of transporters, establishment of cellular interactions, etc), including the suppression of transcytosis as described here, please consider a title that more accurately reflects the results of the study.

We thank the reviewer for pointing out the overgeneralization of the term BBB development, and have amended the title to reflect more specifically what we have measured: “Suppression of transcytosis regulates zebrafish blood-brain barrier function”.

2) The conclusions of the manuscript rely significantly upon the spatiotemporal expression of mfsd2aa (and mfsd2ab to a lesser extent). Based upon the results of this study, it would be predicted that mfsd2aa is not expressed in brain endothelial cells at 3 dpf, which allows for transcytosis of tracers into the brain parenchyma. Furthermore, it would be predicted that mfsd2aa is expressed in brain endothelial cells by 5 dpf, which then suppresses transcytosis of tracers into the brain parenchyma. However, the authors do not provide any expression data for either mfsd2aa or mfsd2ab.In previous studies, Guemez-Gamboa et al. used whole-mount in situ hybridization to show that both zebrafish mfsd2aa or mfsd2ab transcripts are expressed throughout the nervous system in zebrafish at 24, 48, and 96 hpf. However, it is not clear from this data whether brain endothelial cells express either transcript. In addition, Thisse and Thisse showed basal expression of mfsd2aa in spinal cord neurons as early as 20-25 somites to Prim-5 stage, no spatial restriction at the Prim-15 to Prim-25 and High-pec to Long-pec stages, and neurocranium expression at 5 dpf (https://zfin.org/ZDB-GENE-041114-166/expression). Thus, it remains to be determined whether zebrafish mfsd2aa or mfsd2ab is expressed in brain endothelial cells.Please provide expression data that demonstrates mfsd2aa expression (or lack of expression) in zebrafish brain endothelial cells at the appropriate developmental time points (i.e. 3 dpf, 5 dpf, and adult). Perhaps another valid strategy would be to rescue mfsd2aa mutants by cell autonomous expression of wild type mfsd2aa in endothelial cells.Guemez-Gamboa et al. (2015) Nature Genetics 47:809-813Thisse and Thisse, (2004) ZFIN Direct Data Submission (http://zfin.org).

To address this reviewer and reviewer 1’s concern, we have performed fluorescent in situ hybridization (FISH) on sections of 3 and 5 dpf larvae and adult brains from kdrl:mCherry transgenic fish, which fluorescently labels endothelial cells. These FISH experiments have shown that both *mfsd2aa* and *mfsd2ab* are most highly expressed in brain endothelial cells at 5 dpf, While *mfsd2aa* was not expressed above background levels at 3 dpf, *mfsd2ab* was highly expressed at both 3 and 5 dpf (Figure 4). Both paralogues are expressed in the adult brain tissue with higher levels of *mfsd2aa* than *mfsd2ab*. In addition, previously published RNA-seq from FAC-sorted endothelial cells at 48 hpf similarly revealed barely detectable expression of mfsd2aa and high levels of mfsd2ab (Bower et al., Nature Neuroscience, 2014). Taken together, these FISH experiments have nicely supported our hypothesis that the suppression of transcytosis coincides with the onset of *mfsd2aa* expression in the vasculature.

3) To quantify "leakiness", this study focused on the number of parenchymal cells that accumulated tracer in the midbrain and demonstrated that larvae at 3 dpf are leakier than 5 dpf. The authors also indicated that parenchymal cells in the hindbrain did not accumulate tracer at a significant level at either developmental time point. However, in the mfsd2aa mutants, the authors did not examine midbrain leakiness at 3 dpf or in the hindbrain at either 3 dpf or 5 dpf. Do the mfsd2aa mutants at 3 dpf exhibit increased leakiness compared to wild types at 3 dpf? Also, if mfsd2aa suppresses transcytosis in all brain endothelial cells, then is the hindbrain leakier in mfsd2aa mutants at 3 dpf and 5 dpf? If not, please explain.

As stated above, we have altered our method of quantifying leakage throughout the manuscript in response to multiple reviewers’ suggestions. This quantification method has been widely used in mice (Nitta et al., 2003; Bell et al., 2010; Daneman et al., 2010; Andreone et al., 2017; Chow et al., 2017). With this method of normalized brain parenchyma tracer intensity, we have now quantified leakage in *mfsd2aa* and *mfsd2ab* mutants both in the midbrain and hindbrain at 5 dpf to address potential regional differences, or lack thereof. This data is now included in Figure 5. When we examined DBP-EGFP tracer permeability in *mfsd2aa* mutant and wild-type fish at 3 dpf, we did not observe an increase in BBB permeability in *mfsd2aa* mutants compared to the already elevated levels in wild-type fish (data not shown), as expected from the lack of *mfsd2aa* expression at this time (Figure 4).

Reviewer #3:

[…] There are some minor experimental additions that could be performed to strengthen the study's conclusions and impact.1) Given that antibodies to mark tight junctions exist that work in zebrafish, it would be useful to add this analysis to the time course of BBB formation.

We thank the reviewer for her kind remarks. We feel that this information is already available in the Jeong et al., 2008 and the Xie et al., 2010 papers, and therefore do not believe it needs to be repeated. However, we have further highlighted this fact in the Results section.

2) Given the controversy surrounding the timing of BBB closure in zebrafish, it is important to extend the time course to 10 days to be consistent with the Fleming et al. study. Additionally, the hindbrain at 3 days shows ~2 cells/embryo taking up tracer whereas the authors consider the midbrain BBB closed at 5-6 days with ~8 cells/larva taking up tracer. Thus, it is important to test if the number of cells taking up tracer further reduces with time.

We have now repeated the leakage assays at 10 dpf and included that data in Figure 1. Additionally, we have altered our metric of quantifying leakage to be less restricted to the number of parenchymal cells and are measuring tracer intensity in the brain parenchyma instead. This new quantification has been applied to the 10 dpf data.

3) The standard in the field is to rescue a new mutant with wild-type gene product to control for off-target CRISPR/Cas9 effects. Similarly, qPCR for mfsd2aa and mfsd2ab are important controls for the new mutants, especially since phenotypes were not observed in mfsd2ab mutants.

Given the late phenotype (5 dpf) we do not believe that the use of RNA to rescue the *mfsd2aa* leakage will work. While we have not rescued the phenotype, we have generated multiple F0 crispants for both *mfsd2aa* and *mfsd2ab* using new guides to target each gene, and observed similar leakage phenotypes in *mfsd2aa*crispants and did not observe any leakage phenotypes in *mfsd2ab*crispants, suggesting that the observed phenotypes are in fact due to the loss of the target gene (Figure 5—figure supplement 2). Both stable mutant lines have also been backcrossed to wild-type fish for at least 5 generations, and the leakage phenotype in *mfsd2aa* mutants still persists. We have now performed the requested qPCR for *mfsd2aa* and *mfsd2ab* in both mutant backgrounds to ensure reduced transcript levels (Figure 4—figure supplement 1).

*4) Not performing* in vivo *imaging of the mfsd2aa mutants seems like a missed opportunity. Such investigation could also reveal if both types of leakage observed in the time course are affected in mfsd2aa mutants. The analysis presented in Figure 2E would also be helpful to show for the mutants.*

We completely agree with the reviewer that these are critical experiments, and have now performed time lapse leakage assays in *mfsd2aa* mutant and heterozygous siblings, revealing the dynamics of increased transcytosis in the BBB (Figure 6).